# Set-based Neural Network Encoding Without Weight Tying

**Bruno Andreis[1], Soro Bedionita[1], Philip H.S. Torr[2], Sung Ju Hwang[1,3]**

KAIST [1], South Korea
University of Oxford, United Kingdom [2]
DeepAuto.ai, South Korea [3]
`{andries, sorobedio, sjhwang82}@kaist.ac.kr, philip.torr@eng.ox.ac.uk`

## Abstract

We propose a neural network weight encoding method for network property prediction that utilizes set-to-set and set-to-vector functions to efficiently encode neural network parameters. Our approach is capable of encoding neural networks in a model zoo of mixed architecture and different parameter sizes as opposed to previous approaches that require custom encoding models for different architectures. Furthermore, our **S**et-based **N**eural network **E**ncoder (SNE) takes into consideration the hierarchical computational structure of neural networks. To respect symmetries inherent in network weight space, we utilize Logit Invariance to learn the required minimal invariance properties. Additionally, we introduce a *pad-chunk-encode* pipeline to efficiently encode neural network layers that is adjustable to computational and memory constraints. We also introduce two new tasks for neural network property prediction: cross-dataset and cross-architecture. In cross-dataset property prediction, we evaluate how well property predictors generalize across model zoos trained on different datasets but of the same architecture. In cross-architecture property prediction, we evaluate how well property predictors transfer to model zoos of different architecture not seen during training. We show that SNE outperforms the relevant baselines on standard benchmarks.

## 1 Introduction

Recently, deep learning methods have been applied to a wide range of fields and problems. With this broad range of applications, huge volumes of datasets are continually being made available in the public domain together with neural networks trained on these datasets. Given this abundance of trained neural network models, the following curiosity arises: what can we deduce about these networks with access only to the parameter values? More generally, can we predict properties of these networks such as generalization performance on a testset(without access to the test data), the dataset on which the model was trained, the choice of optimizer and learning rate, the number of training epochs, choice of model initialization etc. through an analysis of the model parameters? The ability to infer such fundamental properties of trained neural networks using only the parameter values has the potential to open up new application and research paradigms [De Luigi et al., 2023, Zhou et al., 2023a,b, Navon et al., 2023] such as learning in the latent space of neural network weights for tasks such as weight generation [Schürholt et al., 2021, Soro et al., 2024], latent space transfer of weights across datasets allowing for transferring weights from one dataset to another as was recently demonstrated in Soro et al. [2024] and latent space optimization using gradient descent where optimization is performed on the weight embeddings [Rusu et al., 2018].

We tackle two specific versions of this problem: predicting a) frequencies of Implicit Neural Representations [Sitzmann et al., 2020] (INRs), and b) the performance of CNNs and Transformers,

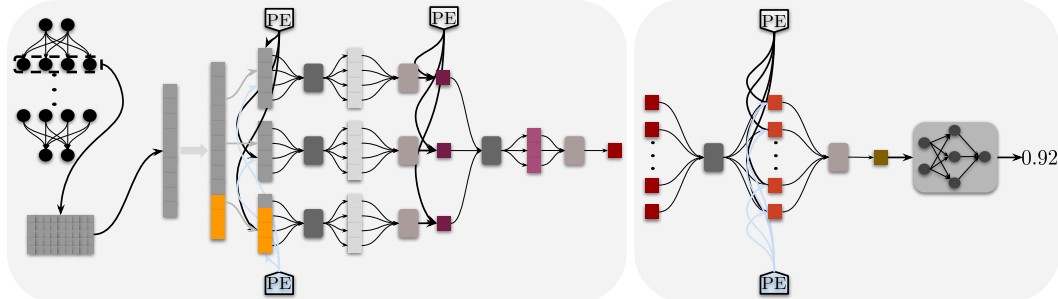

Figure 1: **Legend:** ▮: Padding, ▮: Set-to-Set Function, ▮: Set-to-Vector Function, PE: Layer-Level & PE: Layer-Type Encoder. **Concept:** *(left)* Given layer weights, SNE begins by padding and chunking the weights into *chunksizes*. Each chunk goes through a series of set-to-set and set-to-vector functions to obtain the chunk representation vector. Layer *level* and *type* positional encodings are used to inject structural information of the network at each stage of the chunk encoding process. All chunk encoding vectors are encoded together to obtain the layer encoding. *(right)* All layer encodings in the neural network are encoded to obtain the neural network encoding vector again using as series of set-to-set and set-to-vector functions. This vector is then used to predict the neural network property of interest.

given access only to the network weights. The first approach to solving this problem, proposed by Unterthiner et al. [2020], involves computing statistics such as the mean, standard deviation and quantiles, of each layer in the network, concatenating them to a single vector that represents the neural network encoding, and using this vector to predict the desired network property. Another approach, also proposed as a baseline in Unterthiner et al. [2020], involves flattening all the parameter values of the network into a single vector which is then fed as input to layers of multilayer perceptrons (MLPs) to predict the property of interest. An immediate consequence of this approach is that it is practical only for moderately sized neural network architectures. Additionally, this approach ignores the hierarchical computational structure of neural networks through the weight vectorization process. The second, and most recent approach to this problem, proposed by Zhou et al. [2023a,b], takes a geometric approach to the problem by building neural network weight encoding functions, termed neural functionals, that respect symmetric properties of permutation invariance and equivariance of the hidden layers of MLPs under the action of an appropriately applied permutation group. While this approach respect these fundamental properties in the parameter space, it's application is restricted, strictly, to MLPs. Also, even when relaxations are made to extend this method to convolutional networks and combinations of convolutional layers and MLPs, these only work under strict conditions of equivalence in the channel size in the last convolutional layer and the first linear layer. Hence it is clear that while the methods of Zhou et al. [2023a,b] and Navon et al. [2023] enjoy nice theoretical properties through weight tying, their application is limited to only a small subset of carefully chosen architectures.

Moreover, these approaches [Unterthiner et al., 2020, Zhou et al., 2023a,b, Navon et al., 2023] have a fundamental limitation: their encoding methods are applicable only to a single fixed, pre chosen neural network architecture. Once the performance predictor is trained, in the case of Unterthiner et al. [2020], and the neural network encoder of Zhou et al. [2023a] and Navon et al. [2023] are defined, they cannot be used to predict the performance of neural networks of different architecture. These issues are partly addressed by the graph based approaches of Kofinas et al. [2024] and Lim et al. [2023]. Consequently, evaluating these models on diverse architectures is impossible without training an new performance predictor for each architecture.

To this end, we propose a Set-based Neural Network Encoder (SNE) for property prediction of neural networks given only the model parameters that is agnostic to the network architecture without weight tying. Specifically, we treat the neural network encoding problem from a set encoding perspective by utilizing compositions of *set-to-set* and *set-to-vector* functions. However, the parameters of neural networks are ordered and hierarchical. To retain this order information, we utilize positional encoding [Vaswani et al., 2017] at various stages in our model. Also, our model incorporates the hierarchical computational structure of neural networks in the encoder design by encoding independently, layer-wise, culminating in a final encoding stage that compresses all the layer-wise information into a single encoding vector used to predict the network property of interest. To handle the issue of large and variable parameter sizes efficiently, we incorporate a *pad-chunk-encode* pipeline that is parallelizable and can be used to iteratively encode layer parameters. To learn the correct permutations of MLP weights, we employ the Logit Invariance Regularizer of Moskalev et al. [2023]

instead of weight tying. In terms of evaluation, we introduce two new tasks: cross-dataset neural network property prediction and cross-architecture neural network property prediction. In cross-dataset neural network performance prediction, we fix the network architecture used to generate the training data and evaluate how well the performance predictors transfer to the same architecture trained on different datasets. For cross-architecture neural network performance prediction, we fix only the architecture for generating the training data and evaluate the performance of the predictor on architectures unseen during training. These tasks are important since in the real setting, access to model zoos is scare, making transferability desirable.

Our contributions are as follows:

- We develop a Set-based Neural Network Encoder (SNE, see Figure 1) for network property prediction given access only to parameter values that can encode neural networks of arbitrary architecture, taking into account the hierarchical computational structure of neural networks.

- We introduce the cross-dataset property prediction task where we evaluate how well predictors transfer across neural networks trained on different datasets.

- We introduce the cross-architecture property prediction task where we evaluate how well property predictors trained on a specific architecture transfer to unseen architectures.

- We benchmark SNE against the relevant baselines [Unterthiner et al., 2020, Navon et al., 2023, Zhou et al., 2023a,b] on the cross-dataset task and show significant improvement over the baselines.

- We provide the first set of results on the cross-architecture task, a task for which most of the baselines, with the exception of Zaheer et al. [2017] and Kofinas et al. [2024] under special conditions (see remark in Section 4.2), cannot be used.

## 2 Related Work

**Set Functions:** Neural networks that operate on set (un)structured data have recently been used in many applications ranging from point cloud classification to set generation [Kim et al., 2021]. Set functions are required to respect symmetric properties such as permutation invariance and equivariance. In DeepSets [Zaheer et al., 2017], a set of sum-decomposable functions are introduced that are equivariant in the Set-to-Set applications and invariant in the Set-to-Vector applications. In Set Transformers [Lee et al., 2019], a class of attention based Set-to-Set and Set-to-Vector functions are introduced that are more expressive and capable of modeling pairwise and higher order interactions between set elements. Recent works such as Bruno et al. [2021] and Willette et al. [2023] deal with the problem of processing sets of large cardinality in the the limited memory/computational budget regime. In this work, we utilize the class of set functions developed in Lee et al. [2019] to develop a neural network encoder for performance prediction that is agnostic to specific architectural choices. Our set-based formulation allows us to build such an encoder, capable of handling neural networks weights of arbitrary parameter sizes. This is a deviation from recent approaches to neural network encoding for property prediction that can encode only parameters of a single fixed architecture.

**Neural Network Property Prediction From Weights:** Predicting the properties of neural networks given access only to the trained parameters is a relatively new topic of research introduced by Unterthiner et al. [2020]. In Unterthiner et al. [2020], two methods are proposed for predicting the generalization performance of neural networks: the first involves flattening the weights of the network into a single vector, processing it using multiple layers of MLPs to obtain an encoding vector which is then used to predict the performance. The second involves computing the statistics of each layer in the network, such as mean, variance, quantiles etc., concatenating them into a single vector that is then used for predicting the performance of the network. The most recent approach that we are aware of, Navon et al. [2023] and Zhou et al. [2023a,b], proposes a neural network weight encoder that is invariant or equivariant, depending on the application, to an appropriately applied permutation group to the hidden layers of MLPs. Two variants of their model is provided: one which operates only on the hidden layers, and conforms strictly to the theory of permuting MLP hidden neurons [Hecht-Nielsen, 1990], and a relaxation that assumes that the neurons of both the input and output layers of MLPs are permutable. Additionally, extensions are provided for convolutional layers. Kofinas et al. [2024] and Lim et al. [2023] represent weights using graph neural networks. Our approach, SNE, is directly comparable to these methods on the neural network property prediction task. However,

unlike the methods of Unterthiner et al. [2020], Navon et al. [2023] and Zhou et al. [2023a] which operate only on neural networks of fixed architecture, and consequently fixed number of encodable parameters, SNE is capable of encoding networks of arbitrary architecture. Moreover, SNE utilizes the hierarchical computation structure of neural networks by encoding, iteratively or in parallel, from the input to the output layers. Additionally, we go further than the experimental evaluation in Unterthiner et al. [2020], Navon et al. [2023] and Zhou et al. [2023a,b] by introducing two new tasks: cross-dataset and cross-architecture neural network property prediction. Unterthiner et al. [2020], Navon et al. [2023] and Zhou et al. [2023a,b] can only be benchmarked on the cross-dataset task where all networks in the model zoos are of the same architecture. Their restriction to a single fixed architecture makes cross-architecture evaluation impossible. Our method SNE, and those of Kofinas et al. [2024], Lim et al. [2023], on the other hand can be used for both tasks.

## 3 Set-based Neural Network Encoding Without Weight Tying

### 3.1 Preliminaries

We have access to a dataset $D = \{(x_1, y_1), \ldots, (x_n, y_n)\}$ where for each $(x_i, y_i)$ pair, $x_i$ represents the weights of a neural network architecture $a$, sampled from a set of architectures $\mathcal{A}$ and $y_i$ corresponds to some property of $x_i$ after it has been trained on a specific dataset $d$. $y_i$ can be properties such as generalization gap, training loss, the learning rate used to train $x_i$, or even the number of epochs, choice of weight initialization, and optimizer used to train $x_i$. Henceforth, we refer to $D$ as a *model zoo*. For each $x_i \in D$, $x_i = [w_i^0, \ldots, w_i^{|x_i|}]$ where $w_i^j$ represents the weights (parameters) of the $jth$ layer of the neural network $x_i$, and $|x_i|$ is the total number of layers in $x_i$. Consequently, $w_i^0$ and $w_i^{|x_i|}$ are the input and output layers of $x_i$ respectively. Additionally, we introduce the $\text{Flat} : x_i \rightarrow \mathbf{R}^{d_i}$ operation, that takes as input the weights of a neural network and returns the flattened weights and $d_i$ is the total number of parameter is $x_i$.

The neural network encoding problem is defined such that, we seek to compress $x_i \in \mathbf{R}^{d_i}$ to a compact representation $z_{x_i} \in \mathbf{R}^h$ such that $z_{x_i}$ can be used to predict the properties $y_i$ of $x_i$ with $h \ll d_i$. In what follows, we present the details of our Set-based Neural Network Encoding (SNE) method capable of encoding the weights of neural networks of arbitrary architecture that takes into account the hierarchical computational structure of the given architecture and with efficient methods for processing weights of high dimension.

### 3.2 Handling Large Layer Weights via Chunking

For a given layer $w_i^j \in x_i$, the dimension of $w_i^j$, $|w_i^j|$ can be very large. For instance, when considering linear layers, flattening the weights can results in a tensor that can require large compute memory to be processable by another neural network. To resolve this issue, we resort to *chunking*. Specifically, for all layers $w_i^j \in x_i$, we perform the following operations:

$$\hat{w}_i^j = \text{Chunk}(\text{Pad}(\text{Flat}(w_i^j), c), c) = \{w_i^{j_0}, \ldots, w_i^{j_q}\}, \tag{1}$$

where for any $w_i^{j_t} \in \hat{w}_i^j$, $|w_i^{j_t}| \in \mathbf{R}^c$. Here, $c$ is the *chunksize*, fixed for all layer types in the neural network and $t \in [0, \ldots, q]$. The padding operation $\text{Pad}(w_i^j, c)$ appends zeros, if required, to extend $w_i^j$ and make its dimension a multiple of the chunksize $c$. To distinguish padded values from actual weight values, each element of $\hat{w}_i^j$ has a corresponding set of masks $\hat{m}_i^j = \{m_i^{j_0}, \ldots, m_i^{j_q}\}$. Note that with this padding and subsequent chunking operation, each element in $\hat{w}_i^j$ is now small enough, for an appropriately chosen chunksize $c$, to be processed. Moreover, all the elements in $\hat{w}_i^j$ can be processed in parallel. Importantly, chunksizes are chosen to ensure that weights from the same neurons are grouped together. This allows for principled learning of specific symmetries which we discuss later in Section 3.6. An ablation on the effect of chunksize is provided in Appendix E.

The model zoos we consider in the experimental section are populated by neural networks with stacks of convolutional and linear layers. For each such layer, we apply the padding and chunking operation differently. For a linear layer $w_i^j \in \mathbf{R}^{\text{out} \times \text{in}}$, where out and in are the input and output dimensions respectively, we apply the flattening operation on both dimensions followed by padding and chunking. However for a convolutional layer $w_i^j \in \mathbf{R}^{\text{out} \times \text{in} \times \text{k} \times \text{k}}$, we do not apply the flattening, padding, and

chunking operations to the kernel dimensions k and operate only on the input and output dimensions since the kernels are small enough to be encoded together. Finally we note that for layers with bias values, we apply the procedure detailed above independently to both the weights and biases.

## 3.3  Independent Chunk Encoding

The next stage in our Set-based Neural Network encoding pipeline is the individual encoding of each chunk of weight in $\hat{w}_i^j = \{w_i^{j_0}, \ldots, w_i^{j_t}\}$. For each $w_i^{j_t} \in \hat{w}_i^j$, we treat the $c$ elements as members of a set. However, it is clear that $w_i^{j_t}$ has order in its sequence, *i.e.*, an ordered set. We remedy this by providing this order information via positional encoding. Concretely, for a given $w_i^{j_t} \in \mathbf{R}^{c \times 1}$, we first model the pairwise relations between all $c$ elements using a *set-to-set* function $\Phi_{\theta_1}$ to obtain:

$$\hat{w}_i^{j_t} = \Phi_{\theta_1}(w_i^{j_t}) \in \mathbf{R}^{c \times h}. \tag{2}$$

That is, $\Phi_{\theta_1}$ captures pair-wise correlations in $w_i^{j_t}$ and projects all elements (weight values) to a new dimension $h$.

Given $\hat{w}_i^{j_t} \in \mathcal{R}^{c \times h}$, we inject two kinds of positionally encoded information. The first encodes the *layer type* in a list of layers, *i.e.*, linear or convolution for the model zoos we experiment with, to obtain:

$$\hat{w}_i^{j_t} = \text{PosEnc}_{Layer}^{Type}(\hat{w}_i^{j_t}) \in \mathbf{R}^{c \times h}. \tag{3}$$

Here we abuse notation and assign the output of PosEnc($\cdot$) to $\hat{w}_i^{j_t}$ to convey the fact that $\hat{w}_i^{j_t}$'s are modified in place and to simplify the notation. Also, all PosEnc($\cdot$)s are variants of the positional encoding method introduced in Vaswani et al. [2017]. Next we inject the layer level information. Since neural networks are computationally hierarchical, starting from the input to the output layer, we include this information to distinguish chunks, $w_i^{j_t}$s from different layers. Specifically, we compute:

$$\hat{w}_i^{j_t} = \text{PosEnc}_{Layer}^{Level}(\hat{w}_i^{j_t}) \in \mathbf{R}^{c \times h}, \tag{4}$$

where the input to $\text{PosEnc}_{Layer}^{Level}(\cdot)$ is the output of Equation 3. We note that this approach is different from previous neural network encoding methods [Unterthiner et al., 2020] that loose the layer/type information by directly encoding the entire flattened weights hence disregarding the hierarchical computational structure of neural networks. Experimentally, we find that injecting such positionally encoded information improves the models performance (Ablation E).

We further model pairwise correlations in $\hat{w}_i^{j_t}$, now infused with layer/type information, using another set-to-set function $\Phi_{\theta_2}$:

$$\hat{w}_i^{j_t} = \Phi_{\theta_2}(w_i^{j_t}) \in \mathbf{R}^{c \times h}. \tag{5}$$

The final step in the chunk encoding pipeline involves compressing all $c$ elements in $\hat{w}_i^{j_t}$ to a compact representation. For this, we use a *set-to-vector* function $\Gamma_{\theta_\alpha} : \mathbf{R}^{c \times h} \to \mathbf{R}^h$. In summary, the chunk encoding layer computes the following function:

$$\tilde{w}_i^{j_t} = \Gamma_{\theta_\alpha}[\Phi_{\theta_2}(\text{PosEnc}_{Layer}^{Level}(\text{PosEnc}_{Layer}^{Type}(\Phi_{\theta_1}(w_i^{j_t}))))], \tag{6}$$

where $\tilde{w}_i^{j_t} \in \mathbf{R}^{1 \times H}$. Note that for each chunked layer $\hat{w}_i^j = \{w_i^{j_0}, \ldots, w_i^{j_q}\}$, the chunk encoder, Equation 6, produces a new set $\tilde{w}_i^j = \texttt{Concatenate}[\{\tilde{w}_i^{j_0}, \ldots, \tilde{w}_i^{j_q}\}] \in \mathbf{R}^{q \times h}$, which represents all the encodings of all chunks in a layer.

***Remark*** Our usage of set functions $\Phi_{\theta_1}, \Phi_{\theta_2}$ and $\Gamma_{\theta_\alpha}$ allows us to process layers of arbitrary sizes, which allows us to process neural networks of arbitrary architecture using a single model, a property lacking in previous approaches to neural network encoding [Zhou et al., 2023a,b, Unterthiner et al., 2020, Navon et al., 2023].

## 3.4  Layer Encoding

At this point, we have encoded all the chunked parameters of a given layer to obtain $\tilde{w}_i^j$. Encoding a layer, $w_i^j$, then involves defining a function $\Gamma_{\theta_\beta} : \mathbf{R}^{q \times h} \to \mathbf{R}^{1 \times h}$ for arbitrary $q$. In practice, this is done by computing:

$$\mathbf{w}_i^j = \Gamma_{\theta_\beta}[\text{PosEnc}_{Layer}^{Level}(\Phi_{\theta_3}(\tilde{w}_i^j))] \in \mathbf{R}^{1 \times h}. \tag{7}$$

Again we have injected the layer level information, via positional encoding, into the encoding processed by the set-to-set function $\Phi_{\theta_3}$. We then collect all the layer level encodings of the neural network $x_i$:

$$\tilde{w}_i = \texttt{Concatenate}[\mathbf{w}_i^0, \ldots, \mathbf{w}_i^{|x_i|}] \in \mathbf{R}^{|x_i| \times h}. \tag{8}$$

### 3.5 Neural Network Encoding

With all layers in $x_i$ encoded, we compute the neural network encoding vector $z_{x_i}$ as follows:

$$z_{x_i} = \Gamma_{\theta_\gamma}[\Phi_{\theta_4}(\text{PosEnc}_{Layer}^{Level}(\tilde{w}_i))] \in \mathbf{R}^h. \tag{9}$$

$z_{x_i}$ compresses all the layer-wise information into a compact representation for the downstream task. Since $\Gamma_{\theta_\gamma}$ is agnostic to the number of layers $|x_i|$ of network $x_i$, the encoding mechanism can handle networks of arbitrary layers and by extension architecture. Similar to the layer encoding pipeline, we again re-inject the layer-level information through positional encoding before compressing with $\Gamma_{\theta_\gamma}$.

Henceforth, we refer to the entire encoding pipeline detailed so far as $\text{SNE}_\Theta(x_i)$ for a network $x_i$, where $\Theta$ encapsulates the encoder parameters, $\Phi_{\theta_{1-4}}, \Gamma_\alpha, \Gamma_\beta$ and $\Gamma_\gamma$.

### 3.6 On Minimal Equivariance Without Weight Tying

Given an MLP, there exists permutations of the weights such that the networks are functionally equivalent [Hecht-Nielsen, 1990]. Since not all permutations are functionally correct, the encoder needs to learn the correct functional equivalence. To achieve this, we utilize the concept of Logit Invariance Regularization [Moskalev et al., 2023] where we constrain the output of the non-equivariant $\text{SNE}_\Theta$ (due to the positional encoding of input and output vectors) to respect the restricted functionally correct permutation group. This results in the following optimization problem:

$$\underset{\theta}{\text{minimize}}\, \ell_f(D) + v\ell_f(D, G), \tag{10}$$

where $G$ is the group of functionally equivariant permutations in the weight space, $D$ is the training dataset and $v$ balances the task loss $\ell_f(D)$ and the Logit Invariance Regularization term $\ell_{f(D,G)}$. Proposition 3.1 of Moskalev et al. [2023] guarantees that the resulting SNE model will have low sensitivity to functionally incorrect permutations of the weights. In practice, $\ell_f(D, G)$ is the $L_2$ distance between functionally equivalent permutations of the same weight. This approach differs from previous works [Zhou et al., 2023a,b, Navon et al., 2023] which instead result to weight tying to achieve minimal equivariance.

***Remark*** While we use a regularization approach to achieve the required approximate minimal equivariance in weight-space, our usage of the Logit Invariance Regularizer [Moskalev et al., 2023] theoretically guarantees that we indeed learn the correct invariance property similar to the weight-typing approaches. Additionally, our formulation is what allows us to deal with arbitrary architectures using a single model (see Section 4), as opposed to previous works, since strict enforcement of weight-space equivariance by design requires crafting a new model for different architectures. In this sense, our approach provides a general encoder which in principle is applicable to any architecture, resolving the limitations of purely weight-tying approaches.

Theoretical discussions on the adopted regularization based approach to minimal equivariance versus weight tying is provided in Appendix C.

### 3.7 Choice of Set-to-Set and Set-to-Vector Functions

We specify the choice of Set-to-Set and Set-to-Vector functions encapsulated by $\Phi_{\theta_{1-4}}, \Gamma_\alpha, \Gamma_\beta$ and $\Gamma_\gamma$ used to implement SNE. Let $X \in \mathbf{R}^{n_X \times d}$ and $Y \in \mathbf{R}^{n_Y \times d}$ be arbitrary sets where $n_X = |X|$, $n_Y = |Y|$ and $d$ (note the abuse of notation from Section 3.1 where $d$ is a dataset) is the dimension of an element in both $X$ and $Y$. The MultiHead Attention Block (MAB) with parameter $\omega$ is given by:

$$\text{MAB}(X, Y; \omega) = \text{LayerNorm}(H + \text{rFF}(H)), \text{where} \tag{11}$$

$$H = \text{LayerNorm}(X + \text{MultiHead}(X, Y, Y; \omega)). \tag{12}$$

Here, LayerNorm and rFF are Layer Normalization [Ba et al., 2016] and row-wise feedforward layers respectively. MultiHead$(X, Y, Y; \omega)$ is the multihead attention layer of Vaswani et al. [2017].

The Set Attention Block [Lee et al., 2019], SAB, is given by:

$$\text{SAB}(X) := \text{MAB}(X, X). \tag{13}$$

That is, SAB computes attention between set elements and models pairwise interactions and hence is a Set-to-Set function. Finally, the Pooling MultiHead Attention Block [Lee et al., 2019], $\text{PMA}_k$, is given by:

$$\text{PMA}_k(X) = \text{MAB}(S, \text{rFF}(X)), \quad \text{where} \tag{14}$$

$S \in \mathbf{R}^{k \times d}$ and $X \in \mathbf{R}^{n_X \times d}$. The $k$ elements of $S$ are termed *seed vectors* and when $k = 1$, as is in all our experiments, $\text{PMA}_k$ pools a set of size $n_X$ to a single vector making it a Set-to-Vector function.

All parameters encapsulated by $\Phi_{\theta_{1-4}}$ are implemented as a stack of two SAB modules: $\text{SAB}(\text{SAB}(X))$. Stacking SAB modules enables us not only to model pairwise interactions but also higher order interactions between set elements. Finally, all of $\Gamma_\alpha, \Gamma_\beta$ and $\Gamma_\gamma$ are implemented as a single PMA module with $k = 1$.

### 3.8 Downstream Task

Given $(z_{x_i}, y_i)$, we train a predictor $f_\theta(z_{x_i})$ to estimate properties of interest of the network $x_i$. In this work, we focus solely on the task of predicting the generalization performance of $x_i$, where $y_i$ is the performance on the test set of the dataset used to train $x_i$ for CNNs and frequencies for INRs. The parameters of the predictor $f_\theta$ and all the parameters in the neural network encoding pipeline, $\Theta$, are jointly optimized. In particular, we minimize the error between $f_\theta(z_{x_i})$ and $y_i$. For a model zoo, the objective is given as:

$$\underset{\Theta, \theta}{\text{minimize}} \sum_{i=1}^{d} \ell[f_\theta(\text{SNE}_\Theta(x_i)), y_i], \tag{15}$$

for an appropriately chosen loss function $\ell(\cdot)$. In our experiments, $\ell(\cdot)$ is the binary cross entropy or mean squared error loss. The entire SNE pipeline is shown in Figure 1.

## 4 Experiments

We present experimental results on INRs, and the standard CNN benchmark model zoos used in Unterthiner et al. [2020],Zhou et al. [2023a],Zhou et al. [2023b], and Navon et al. [2023]. Experimental settings, hyperparameters, model specification, ablation of SNE and discussions on applying SNE to architectures with branches (*e.g.* ResNets) in Appendix D.

**Baselines:** We compare SNE with the following baselines: **a) MLP:** This model flattens the entire weight of the network and encodes it using a stack of MLP layers. **b) DeepSets:** [Zaheer et al., 2017] This model treats the weights as a set with no ordering. **c) HyperRep:** [Schürholt et al., 2021] This model learns a generative model of the flattened weight vector. **d) STATNN** [Unterthiner et al., 2020]**:** This model computes the statistics of each layer such as the mean, variance and quantiles, and concatenates them to obtain the neural network encoding vector. **e) DWSNet** [Navon et al., 2023]**:** is a minimally equivariant model using weight tying developed mainly for MLPs. Various modules are provided for encoding biases, weights and combinations of these two. **f) NFN**$_\text{HNP}$**, NFN**$_\text{NP}$ and **NFT** [Zhou et al., 2023a,b]**:** These models, termed Neural Functionals(NF), are developed mainly for MLPs and utilize weight tying to achieve minimal equivariance. HNP, hidden neural permutation, is applied only to the hidden layers of each network since the output and input layers of MLPs are not invariant/equivariant to the action of a permutation group on the neurons. NP, neural permutation, makes a strong assumption that both the input and output layers are also invariant/equivariant under the action of a permutation group. NFT is similar to both models and utilizes attention layers. **g) NeuralGraph:** [Kofinas et al., 2024] This method represents the weights of a network as a graph and uses graph pooling techniques to obtain the network representation vector. We note that we are unable to benchmark against **Graph Metanetworks** [Lim et al., 2023], which uses a graph approach similar to NeuralGraph, since no code or data is publicly available.

In all Tables, the best methods are shown in **red** and the second in blue. Additionally an extensive ablation of all the components of SNE is provided in Appendix E. All experiments are performed with a single GeForce GTX 1080 TI GPU with 11GB of memory.

## 4.1 Encoding Implicit Neural Representations

**Dataset and Network Architecture:** We utilize the model zoo of Navon et al. [2023] consisting of INRs [Sitzmann et al., 2020] fit to sine waves on $[-\pi, \pi]$ with frequencies sampled from $U(0.5, 10)$. INRs are neural parameterizations of signals such as images using multi-layer perceptrons.

**Task:** The goal is to predict the frequency of a given INR. Each INR is encoded to a 32 dimensional vector which is then fed to a classifier with two linear layers of dimension 512.

Table 1: Predicting Frequencies of Implicit Neural Representations (INRs).

| Model | #Params | MSE |
|---|---|---|
| MLP | 14K | $1.917_{\pm 0.241}$ |
| Deepsets | 99K | $2.674_{\pm 0.740}$ |
| STATNN | 44K | $0.937_{\pm 0.276}$ |
| $\text{NFN}_{\text{NP}}$ | 2.0M | $0.911_{\pm 0.218}$ |
| $\text{NFN}_{\text{HNP}}$ | 2.8M | $0.998_{\pm 0.382}$ |
| NFT | 6M | $0.401_{\pm 0.109}$ |
| DWSNet | 1.5M | $0.209_{\pm 0.026}$ |
| SNE(Ours) | 358K | $\mathbf{0.098}_{\pm 0.002}$ |

**Results:** As can be seen in Table 1, SNE significantly outperforms the baselines on this task. Given that INRs are MLPs, minimal equivariance is particularly important for this task and shows that SNE learns the correct minimal equivariance required to solve the task using the logit invariance approach. Additionally, compared to the minimal equivariance constrained models [Zhou et al., 2023a,b, Navon et al., 2023], SNE is parameter efficient as show in Table 1. We note that increasing the parameter counts of the MLP, DeepSets and STATNN baselines results in overfitting and poor performance.

## 4.2 Cross-Architecture Performance Prediction

For this task, we train the encoder on 3 homogeneous model zoos of the same architecture and test on 3 homogeneous model zoos of a different architecture unseen during training. The cross-architecture task demonstrates the encoder's agnosticism to particular architectural choices since training and testing are done on model zoos of different architectures, *i.e.*, we perform out-of-distribution evaluation.

Table 2: Cross-Architecture Performance Prediction.

| $\text{Arch}_1 \rightarrow \text{Arch}_2$ | DeepSets | NeuralGraph | SNE(Ours) |
|---|---|---|---|
| MNIST→ MNIST | $0.460_{\pm 0.001}$ | $0.473_{\pm 0.079}$ | $\mathbf{0.490}_{\pm 0.027}$ |
| MNIST→ CIFAR10 | $0.508_{\pm 0.001}$ | $0.528_{\pm 0.065}$ | $\mathbf{0.586}_{\pm 0.036}$ |
| MNIST→ SVHN | $0.546_{\pm 0.001}$ | $0.502_{\pm 0.119}$ | $\mathbf{0.535}_{\pm 0.004}$ |
| CIFAR10→CIFAR10 | $0.507_{\pm 0.000}$ | $0.463_{\pm 0.131}$ | $\mathbf{0.660}_{\pm 0.016}$ |
| CIFAR10→MNIST | $0.459_{\pm 0.000}$ | $0.352_{\pm 0.104}$ | $\mathbf{0.558}_{\pm 0.037}$ |
| CIFAR10→SVHN | $0.545_{\pm 0.000}$ | $0.534_{\pm 0.123}$ | $\mathbf{0.581}_{\pm 0.024}$ |
| SVHN→SVHN | $0.553_{\pm 0.000}$ | $0.573_{\pm 0.067}$ | $\mathbf{0.609}_{\pm 0.039}$ |
| SVHN→MNIST | $0.480_{\pm 0.001}$ | $0.448_{\pm 0.044}$ | $\mathbf{0.531}_{\pm 0.039}$ |
| SVHN→CIFAR10 | $0.529_{\pm 0.000}$ | $0.539_{\pm 0.057}$ | $\mathbf{0.622}_{\pm 0.057}$ |

**Datasets and Neural Network Architectures:**
We utilize model zoos trained on MNIST, CIFAR10 and SVHN datasets. We generate a model zoo for these dataset with an architecture consisting of 3 convolutional layers followed by two linear layers and term the resulting model zoo $\text{Arch}_1$. Exact architectural specifications are detailed in Appendix G. We generate the model zoos of $\text{Arch}_2$ following the routine described in Appendix A.2 of Unterthiner et al. [2020]. We refer to the model zoos of Unterthiner et al. [2020] as $\text{Arch}_2$. All model zoos of $\text{Arch}_1$ are used for training and those of $\text{Arch}_2$ are used for testing and are *not* seen during training.

**Task:** Here, we seek to explore the following question: Do neural network performance predictors trained on model zoos of $\text{Arch}_1$ transfer or generalize to the out-of-distribution model zoos of $\text{Arch}_2$? Additionally, we perform cross-dataset evaluation on this task where cross evaluation is with respect to model zoos of $\text{Arch}_2$, *i.e.*, we also check for out-of-distribution transfer across datasets.

**Baselines:** We compare SNE with DeepSets [Zaheer et al., 2017] and NeuralGraph [Kofinas et al., 2024] for this task. None of the other baselines, MLP, STATNN [Unterthiner et al., 2020], $\text{NFN}_{\text{NP}}$ [Zhou et al., 2023a], $\text{NFN}_{\text{HNP}}$ [Zhou et al., 2023a], NFT [Zhou et al., 2023b] and DWS-Net [Navon et al., 2023] can be used for this task since they impose architectural homogeneity and hence cannot be used for out-of-distribution architectures by design.

**Results:** We report the quantitative evaluation on the cross-architecture task in Table 2 and report Kendall's $\tau$ [Kendall, 1938]. The first column, $\text{Arch}_1 \rightarrow \text{Arch}_2$ shows the direction of transfer, where we train using model zoos of $\text{Arch}_1$ and test on model zoos of $\text{Arch}_2$. Additionally, A→B, *e.g.* MNIST→CIFAR10 shows the cross-dataset transfer. From Table 2, it can be seen that SNE transfers best across out-of-distribution architectures and datasets outperforming the DeepSets and NeuralGraph baselines significantly. Interestingly, the DeepSets model, which treats the entire weight

as set with no ordering performs better than the Neu-ralGraph model on average for this task. In conclusion, SNE shows strong transfer across architecture and datasets in the out-of-distribution benchmark.

Table 3: Cross-Architecture Performance Prediction on Schürholt et al. [2022]'s model zoo.

| Dataset | HyperRep | SNE(Ours) |
|---|---|---|
| SVHN → SVHN | 0.45 | **0.67** |
| SVHN → MNIST | 0.15 | **0.61** |
| SVHN → CIFAR10 | 0.10 | **0.68** |

***Remark:*** In Table 2, we performed $Arch_1 \rightarrow Arch_2$ evaluation specifically to allow benchmarking against NeuralGraph [Kofinas et al., 2024]. Specifically, the convolutional filters of $Arch_1$ are larger than those of $Arch_2$. NeuralGraph requires specifying this maximum filter size before training and can only be used to transfer across architectures with filter sizes equal or smaller than the predefined filter size (this can be verified from the official source code[1]), *i.e.*, NeuralGraph is not truly agnostic to architectural choices. SNE on the other hand does not have this limitation. To demonstrate this, we use the SNE model trained in Table 2 and test it on the SVHN model zoo of Schürholt et al. [2022] which is a much larger architecture with larger filter sizes than those of $Arch_1$. We compare with HyperRep [Schürholt et al., 2021] which is trained fully on the testing model zoo. We emphasize that SNE is *not* trained on the training set of this model zoo. Results for HyperRep are taken from Schürholt et al. [2022] and a single SNE model is evaluated to match the setting of Schürholt et al. [2022]. From Table 3, SNE significantly outperforms HyperRep by very large margins without being trained on the training set of Schürholt et al. [2022] as HyperRep and demonstrates true agnosticism to architectural choices compared to NeuralGraph. We report Kendall's $\tau$ [Kendall, 1938] in Table 3.

**Evaluation on Transformers:** We generate a model zoo of transformer using PytorchViT [2024] and test the transfer from $Arch_1$ to the transformer model zoo. For this task, we are unable to benchmark against NeuralGraph as was done in Table 2, since the model cannot process transformer weights when trained on $Arch_1$. Hence we benchmark against the DeepSets baseline as in Table 2. From Table 4 SNE generalizes better to the unseen transformer architecture

Table 4: Cross-Architecture Performance on Transformers. We report Kendall's $\tau$.

| $Arch_1 \rightarrow$ Transformer | DeepSets | SNE(Ours) |
|---|---|---|
| MNIST → MNIST | $0.1975 \pm 0.000$ | $\mathbf{0.4625} \pm_{0.006}$ |
| CIFAR10 → MNIST | $0.1970 \pm 0.000$ | $\mathbf{0.3278} \pm_{0.029}$ |
| SVHN → MNIST | $0.1906 \pm 0.000$ | $\mathbf{0.3735} \pm_{0.009}$ |

at test time than the baselines showing strong architectural transfer. Additionally, here, the model encodes an architecture with about 5 times the number of parameters in $Arch_1$ demonstrating the scalability of our approach. The DeepSets baseline fails to generalize on this task.

## 4.3 Cross-Dataset Performance Prediction

For this task, we train neural network performance predictors on 4 homogeneous model zoos, of the same architecture, with each model zoo restricted to a single dataset.

Table 5: Cross-Dataset Prediction. We report Kendall's $\tau$.

| Dataset | MLP | STATNN | NFN$_{NP}$ | NFN$_{HNP}$ | SNE(Ours) |
|---|---|---|---|---|---|
| MNIST | $0.618_{\pm0.177}$ | $0.788_{\pm0.097}$ | $0.780_{\pm0.107}$ | $0.775_{\pm0.115}$ | $\mathbf{0.807}_{\pm0.094}$ |
| FashionMNIST | $0.613_{\pm0.176}$ | $0.696_{\pm0.170}$ | $\mathbf{0.768}_{\pm0.110}$ | $0.727_{\pm0.142}$ | $0.765_{\pm0.114}$ |
| CIFAR10 | $0.576_{\pm0.062}$ | $0.743_{\pm0.117}$ | $0.731_{\pm0.131}$ | $0.680_{\pm0.177}$ | $\mathbf{0.743}_{\pm0.133}$ |
| SVHN | $0.604_{\pm0.137}$ | $0.709_{\pm0.107}$ | $0.705_{\pm0.120}$ | $0.638_{\pm0.163}$ | $\mathbf{0.730}_{\pm0.100}$ |

**Datasets and Network Architecture:** Each model zoo is trained on one of the following datasets: MNIST [Deng, 2012], FashionMNIST [Xiao et al., 2017], CIFAR10 [Krizhevsky, 2009] and SVHN [Netzer et al., 2018]. We use the model zoos of Unterthiner et al. [2020].

A thorough description of the model zoo generation process can be found in Appendix A.2 of Unterthiner et al. [2020].

**Task:** We train network encoders on a single model zoo, *e.g.* MNIST, and evaluate it on the test set of all four datasets and report the averaged performance. This results in in-distribution testing with respect to the dataset used for training and out-of-distrution testing with respect to the other three datasets, *e.g.* a model trained on MNIST is tested on MNIST, FashionMNIST, CIFAR10 and SVHN.

**Baselines:** We benchmark against MLP, STATNN [Unterthiner et al., 2020], NFN$_{HP}$ [Zhou et al., 2023a], and the NFN$_{HNP}$ [Zhou et al., 2023a].

**Results:** We present the results for this task in Table 5. Here we see that SNE again performs better than the competing methods significantly demonstrating strong transfer across different datasets for the same architecture.

---

[1]https://github.com/mkofinas/neural-graphs

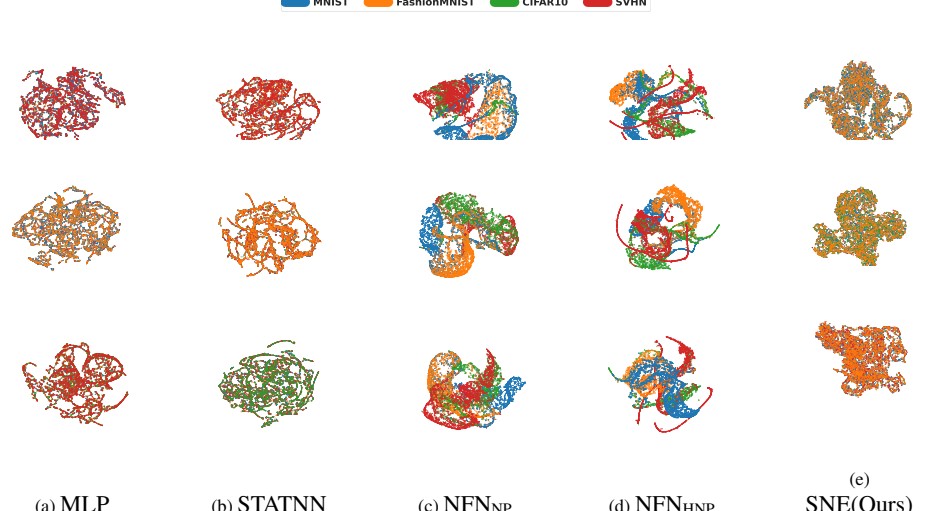

Figure 2: TSNE Visualization of Neural Network Encodings. We train neural network performance prediction methods on a combination of the MNIST, FashionMNIST, CIFAR10 and SVHN modelzoos of Unterthiner et al. [2020]. We present 3 views of the resulting 3-D plots showing how neural networks from each modelzoo are embedded/encoded by the corresponding models. Larger versions of these figures are provided in Appendix K. Zoom in for better viewing.

**Qualitative Analysis:** To understand how SNE transfers well across model zoos, we generate TSNE [Van der Maaten and Hinton, 2008] plots for the neural network encodings of all benchmarked methods on all four homogeneous model zoos in Figure 2. We provide 3 different views of each models embeddings to better illustrate the encoding pattern. In Figures 2c and 2d, we observe that NFN$_{NP}$ and NFN$_{HNP}$ have very clear separation boundaries between the networks from each model zoo. In Figures 2a and 2b, MLP and STATNN, respectively show similar patterns with small continuous strings of model zoo specific groupings. However, these separations are not as defined as those of NFN$_{NP}$ and NFN$_{HNP}$. The embedding pattern of SNE on the other hand is completely different. In Figure 2e, all networks from all the model zoos are embedded almost uniformly close to each other. This may suggest why SNE performs much better on the cross-dataset performance prediction task since it is much easier to interpolate between the neural network encodings generated by SNE across model zoos.

## 5 Conclusion

In this work, we tackled the problem of encoding neural networks for property prediction given access only to trained parameter values. We presented a Set-based Neural Network Encoder (SNE) that reformulates the neural network encoding problem as a set encoding problem. Using a sequence of set-to-set and set-to-vector functions, SNE utilizes a pad-chunk-encode pipeline to encode each network layer independently; a sequence of operations that is parallelizable across chunked layer parameter values. SNE also utilizes the computational structure of neural networks by injecting positionally encoder layer type/level information in the encoding pipeline. As a result, SNE is capable of encoding neural networks of different architectures as opposed to previous methods that only work on a fixed architecture. To learn the correct minimal equivariance for MLP weight permutations, we utilized Logit Invariance Regularization as opposed to weight tying used in previous methods. Experimentally, we introduced the cross-dataset and cross-architecture neural network property prediction tasks. We demonstrated SNE's ability to transfer well across model zoos of the same architecture but with networks trained on different datasets on the cross-dataset task. On the cross-architecture task, we demonstrated SNE's agnosticism to architectural choices and provided the first set of experimental results for this task that demonstrates transferability across architectures.

## Acknowledgments and Disclosure of Funding

This work was supported by Institute for Information & communications Technology Promotion(IITP) grant funded by the Korea government(MSIP) (No.2019-0-00075 Artificial Intelligence Graduate School Program(KAIST)), by Samsung Research Funding Center of Samsung Electronics (No. IO201210-08006-01), Institute of Information & communications Technology Planning & Evaluation (IITP) under Open RAN Education and Training Program (IITP-2024-RS-2024-00429088) grant funded by the Korea government(MSIT), by Google Research Grant and by the UKRI grant: Turing AI Fellowship EP/W002981/1. We would also like to thank the Royal Academy of Engineering and FiveAI.

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

# A    Organization

In Section B we provide some limitations and future directions for the neural network encoding task. In Section C, we discuss weight tying versus regularization approaches to achieving minimal equivariance for MLPs together with some theoretical considerations. In Section D, we discuss how SNE can be applied to architectures with branches, specifically ResNets [He et al., 2016]. In Section E, we provide an ablation on the various components of the proposed method. In Section F, we provide additional experimental results on the cross-dataset and cross-architecture tasks. In Section G, we specify all the model architectures used for generating the model zoos of Arch$_1$ and Arch$_2$ used for the cross-dataset and cross-architecture tasks. In Section H we provide details of the train/validation/test splits. In Section I, we detail all the hyperparameters used for all experiments. We provide implementation details for SNE in Section J and enlarged versions of Figure 2 in Section K.

# B    Limitations & Future Work

In this work, we focused solely on the task of predicting the properties, specifically generalization performance, of neural networks given access only to the model parameters. While the task of encoding neural network weights is a relatively new topic of research with very few baselines, we anticipate new applications/research directions where the neural network encoding vector is used for tasks such as neural network generation or neural network retrieval. These tasks, potential applications, and consideration of more complicated neural network architectures (see discussion in Section D) are out of the scope of this paper and we leave it for future works.

Additionally, we have not included results on larger architectures such as Transformers for the following reasons: Firstly, we are unaware of any such extensive model zoos for such architectures. In Lim et al. [2023], such a model zoo is evaluated, however, this is not publicly available hence we are unable to benchmark against such architectures. Lastly, given that all experiments in this paper were carried out on a single GPU with 11GB of memory, we are unable to generate the extensive Transformer model zoos used in Lim et al. [2023]. However in Section D, we discuss how these architectures can be encoded with our method when such model zoos become available.

# C    Weight Tying vs Regularization and Theoretical Considerations

With regards to achieving minimal equivariance: respecting only a subset of permutations in the weight space that are functionally equivalent, there are two approaches to satisfying this property.

**Weight Tying:** This approach is adopted by Zhou et al. [2023a,b], Navon et al. [2023] where minimal equivariance is baked into the neural network encoder, *i.e.*, weight tying. However, this has a couple of drawbacks:

- It requires a specification of the neural network to be encoded beforehand. This results in encoders that cannot be applied to different architectures as we demonstrated in Section 4.2 where at test time, we evaluate SNE on architectures not seen during training.

- It requires processing all layers of the network together as opposed to the layer-wise encoding scheme that we adopt. Note that for large networks, Zhou et al. [2023a,b], Navon et al. [2023] will require correspondingly large memory but the memory requirements of SNE remains constant since we share the same encoder across all layers, allowing our model to process arbitrary network architectures efficiently.

**Regularization:** Recent works [Moskalev et al., 2023, Cohen-Karlik et al., 2020, Kim et al., 2023, Miyato et al., 2022, Otto et al., 2023] have shown that symmetric constraints such as permutation invariance/equivariance of a non-symmetric model can be achieved through regularization during training. In this work, we take this approach since it offers the following useful properties:

- The level of symmetric constraints, *i.e.*, minimal equivariance, can be learned directly from data without architectural constraints.

- It allows us to process layers independently, resulting in a model that is general and disentangled from architectural choices as we demonstrated in the cross architecture experiments.

In the Ablation study E, we show that the regularization approach indeed helps to learn a model that generalizes well and without it, the model struggles to learn the correct minimal equivariance.

**Theoretical Considerations** A natural question that arises is how well the regularization approach learns the correct symmetric properties of interest. We first define the Logit Invariance, from which we obtain the Logit Invariance Regularization loss of Moskalev et al. [2023].

**Definition C.1 (Logit Invariance)** *[Moskalev et al., 2023] For a given group $G$, dataset $D$ and non-symmetric(with respect to $G$) model $f$, the Logit Invariance Loss of $f$ is given by:*

$$L(D, G) = \sum_{x \sim D} \sum_{g \sim G} \frac{1}{2} ||f(x) - f(gx)||_2^2. \tag{16}$$

Logit Invariance measures the sensitivity of $f$ to the action of $G$ on data samples in $D$.

The following proposition of Moskalev et al. [2023] characterizes the performance decay of the Logit Invariance Regularizer under some linearity assumptions.

**Proposition C.1 (Invariance-Induced by Spectra Decay)** *[Moskalev et al., 2023] Logit invariance error minimization implies $\sigma_{max}(W(t)) \leq \sigma(W(0))$ when $t \to \infty$.*

*proof:* See Section 3.3 of Moskalev et al. [2023].

Here, $W$ are model parameters, $t$ optimization steps and $\sigma_{max}(W) = ||W||_2$. Proposition C.1, through an analysis of the training dynamics of $f$ asserts that $f$ will be insensitive to actions of $G$ on data instances thereby learning the correct invariance property when the Logit Invariance Regularizer is minimized.

In summary, the SNE model constrained by the Logit Invariance Regularizer, learns the correct minimal equivariance required for processing MLP weights without having to resort to weight tying as the methods of Navon et al. [2023] and Zhou et al. [2023a,b].

## D   Considering Architectures with Branches

Here, we outline how SNE may be applied to a model zoo of architectures such as ResNets where branches exist in the computational graph. Given that residual blocks are composed of convolutional and linear layers, each of these can be encoded independently as we already do. To account for residual connections, we propose to introduce special tokens (just as was done for layer types) when we encode the entire block. Additionally, any symmetries inherent in such blocks can be enforced using the invariance regularization term introduced. Given that Transformers are composed of linear layers, the same pipeline can be applied to encode transformers and logit invariance regularization can be used to respect their inherent symmetries.

While we do not see any impediment to applying SNE to such architectures, the unavailability of such model zoos makes it impossible to experimentally verify these and hence we leave it as future work when such model zoos become publicly available.

## E   Ablation

Table 6: Ablation on SNE Components

| Model | MSE |
|---|---|
| W/O Layer Level Encoder | 2.216±1.303 |
| W/O Layer Type Encoder | 0.128±0.007 |
| W/O Set Functions | 1.931±0.108 |
| W/O Positional & Hierarchical Encoding | 7.452±0.799 |
| W/O Invariance Regularization | 0.156±0.015 |
| SNE | **0.098±0.002** |

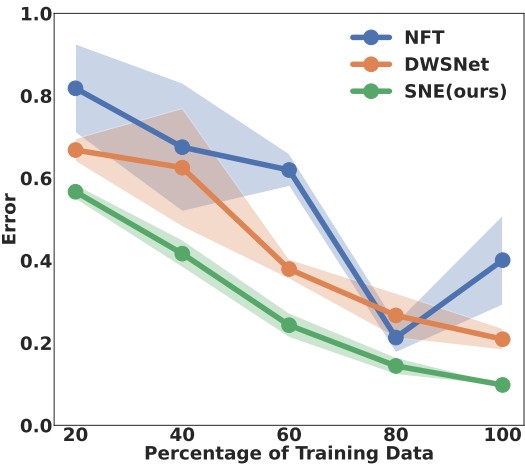

Figure 3: **Ablation:** We compare the performance of models in a limited training data setting using the experiments of Table 1. As shown, SNE is more data efficient than the baseline models when the amount of training data is constrained.

### E.1  Components of SNE

We investigate the impact of the invariant regularization loss, the hierarchical and positional encoding modules, the usage of set functions, layer level and layer type encoders on the performance of SNE using the INR dataset. In Table 6, it can be seen that the hierarchical and positional encoding modules play an important role in the performance of SNE. Removing this module results in significant performance degradation from 0.098 to 7.452. Without positional encoding, SNE is fully permutation equivariant even to functionally incorrect permutations of the weights, making it difficult for the model to learn the correct restricted set of permutations. Secondly, removing the invariance regularization term results in a degradation in performance, from 0.098 to 0.160, as it also makes the task learning the correct minimal equivariance subgroup difficult. We find this only to have a huge impact for MLPs such as INRs where weight permutations are well defined. Additionally, removing the set functions and the layer level and type encoders from the model results in performance drop.

### E.2  Effect of Chunksize

Table 7: Effect of Chunksize.

| Chunksize | MSE |
|-----------|-----|
| 4 | $0.095 \pm 0.012$ |
| 8 | $0.090 \pm 0.023$ |
| 16 | $0.118 \pm 0.019$ |
| 32 | $0.056 \pm 0.020$ |

We provide ablation on the effect of chunksize in the table below for the INR experiment presented in Table 1 of the main paper. From Table 7 we see that performance stays almost the same until the largest chunksize. Hence the chunksize is selected to suit the memory requirements.

### E.3  How Well Is The Logit Invariance Regularization Term Minimized?

We compute the logit invariance loss for the experiments for 5 functionally equivalent permutations of INR weights and obtain a loss of approximately $10^{-15}$. This low loss implies that the SNE learns the correct level of invariance which is the desired property of the non-weight tying approach that we take.

Table 8: Cross-Dataset Neural Network Performance Prediction. We benchmark how well each method transfers across multiple datasets. In the first column, $A \rightarrow B$ implies that a model trained on a homogeneous model zoo of dataset $A$ is evaluated on a homogeneous model zoo of dataset $B$. In the last row, we report the averaged performance of all methods across the cross-dataset task. For each row, the best model is shown in red and the second best in blue. Models are evaluated in terms of *Kendall's* $\tau$, a rank correlation measure.

| | MLP | STATNN | NFN$_{NP}$ | NFN$_{HNP}$ | SNE(ours) |
|---|---|---|---|---|---|
| MNIST$\rightarrow$ MNIST | 0.878±0.001 | 0.926±0.000 | 0.937±0.000 | 0.942±0.001 | 0.941±0.000 |
| MNIST$\rightarrow$ FashionMNIST | 0.486±0.019 | 0.756±0.006 | 0.726±0.005 | 0.690±0.008 | 0.773±0.009 |
| MNIST$\rightarrow$ CIFAR10 | 0.562±0.024 | 0.773±0.005 | 0.756±0.010 | 0.758±0.000 | 0.792±0.008 |
| MNIST$\rightarrow$ SVHN | 0.544±0.005 | 0.698±0.005 | 0.702±0.005 | 0.710±0.010 | 0.721±0.001 |
| FashionMNIST$\rightarrow$ FashionMNIST | 0.874±0.001 | 0.915±0.000 | 0.922±0.001 | 0.935±0.000 | 0.928±0.001 |
| FashionMNIST$\rightarrow$ MNIST | 0.507±0.007 | 0.667±0.010 | 0.755±0.018 | 0.617±0.012 | 0.722±0.005 |
| FashionMNIST$\rightarrow$ CIFAR10 | 0.515±0.007 | 0.698±0.029 | 0.733±0.007 | 0.695±0.032 | 0.745±0.008 |
| FashionMNIST$\rightarrow$ SVHN | 0.554±0.006 | 0.502±0.043 | 0.663±0.014 | 0.662±0.003 | 0.664±0.003 |
| CIFAR10$\rightarrow$ CIFAR10 | 0.880±0.000 | 0.912±0.001 | 0.924±0.002 | 0.931±0.000 | 0.927±0.000 |
| CIFAR10$\rightarrow$ MNIST | 0.552±0.003 | 0.656±0.005 | 0.674±0.018 | 0.600±0.025 | 0.648±0.006 |
| CIFAR10$\rightarrow$ FashionMNIST | 0.514±0.005 | 0.677±0.004 | 0.629±0.031 | 0.526±0.038 | 0.643±0.006 |
| CIFAR10$\rightarrow$ SVHN | 0.578±0.005 | 0.728±0.004 | 0.697±0.006 | 0.662±0.004 | 0.753±0.007 |
| SVHN$\rightarrow$ SVHN | 0.809±0.003 | 0.844±0.000 | 0.855±0.001 | 0.862±0.002 | 0.858±0.003 |
| SVHN$\rightarrow$ MNIST | 0.545±0.025 | 0.630±0.009 | 0.674±0.008 | 0.647±0.016 | 0.647±0.001 |
| SVHN$\rightarrow$ FashionMNIST | 0.523±0.026 | 0.616±0.007 | 0.567±0.014 | 0.494±0.023 | 0.655±0.003 |
| SVHN$\rightarrow$ CIFAR10 | 0.540±0.027 | 0.746±0.002 | 0.725±0.007 | 0.547±0.039 | 0.760±0.006 |
| Cross-Dataset Task | 0.616±0.143 | 0.734±0.115 | 0.746±0.106 | 0.705±0.140 | **0.761±0.101** |

## E.4    Measuring the Importance of Training Set Size

We provide a plot of training set size versus error in Figure 3 using the INR experiment presented in Table 1. From this it can be seen that SNE is more data efficient compared to the baselines [Navon et al., 2023, Zhou et al., 2023b] across all percentages of the full training data demonstrating that the proposed method learns a good embedding even in the limited data setting.

# F    Additional Experimental Results

In this section we provide additional experimental results on the cross-dataset and cross-architecture tasks.

## F.1    Cross-Dataset Evaluation

For this task, we train neural network performance predictors on 4 homogeneous model zoos, of the same architecture, with each model zoo restricted to a single dataset.

**Datasets and Network Architecture:** Each model zoo is trained on one of the following datasets: MNIST [Deng, 2012], FashionMNIST [Xiao et al., 2017], CIFAR10 [Krizhevsky, 2009] and SVHN [Netzer et al., 2018]. We use the model zoos provided by Unterthiner et al. [2020]. To create each model zoo, 30K different hyperparameter configurations were sampled. The hyperparameters include the learning rate, regularization coefficient, dropout rate, the variance and choice of initialization, activation functions etc. A thorough description of the model zoo generation process can be found in Appendix A.2 of Unterthiner et al. [2020]. Architectural descriptions for the model zoos are outlined in Appendix G. Each model zoo is split into a training, testing and validation splits.

**Task:** In this task, we consider cross-dataset neural network performance prediction where we evaluate the prediction performance on the testset of the model zoo on which the predictors were trained on. Additionally, we evaluate how well each predictor transfers to the other model zoos. We evaluate all models using Kendall's $\tau$ [Kendall, 1938].

**Results:** We present the results of the cross-dataset performance prediction task in Table 8. For each row in Table 8, the first column shows the cross-dataset evaluation direction. For instance, MNIST$\rightarrow$CIFAR10 implies that a model trained on an MNIST model zoo is cross evaluated on a model zoo populated by neural networks trained on CIFAR10. We note that the A$\rightarrow$A setting, *e.g.* MNIST$\rightarrow$MNIST, corresponds to the evaluation settings of Unterthiner et al. [2020] and Zhou et al. [2023a]. Also in Table 8 we show the best model in red and the second best model in blue.

Table 9: Cross-Architecture NN Performance Prediction. We show how SNE transfers across architectures and report Kendall's $\tau$.

| Arch$_2$ → Arch$_1$ | SNE |
|---|---|
| MNIST→ MNIST | $0.452 \pm 0.021$ |
| MNIST→ CIFAR10 | $0.478 \pm 0.010$ |
| MNIST→ SVHN | $0.582 \pm 0.016$ |
| CIFAR10→ CIFAR10 | $0.511 \pm 0.020$ |
| CIFAR10→ MNIST | $0.467 \pm 0.020$ |
| CIFAR10→ SVHN | $0.594 \pm 0.029$ |
| SVHN→ SVHN | $0.621 \pm 0.013$ |
| SVHN→ MNIST | $0.418 \pm 0.096$ |
| SVHN→ CIFAR10 | $0.481 \pm 0.055$ |

Table 10: Arch$_1$ for MNIST, FashionMNIST, CIFAR10 and SVHN.

| Output Size | Layers |
|---|---|
| $1 \times 32 \times 32$ | `Input Image` |
| $16 \times 30 \times 30$ | `Conv2d(in_channels=1 , out_channels=16, kernel_size=3), ReLU` |
| $16 \times 28 \times 28$ | `Conv2d(in_channels=16, out_channels=16, kernel_size=3), ReLU` |
| $16 \times 26 \times 26$ | `Conv2d(in_channels=16, out_channels=16, kernel_size=3), ReLU` |
| $16 \times 1 \times 1$ | `AdaptiveAvgPool2d(output_size=(1, 1))` |
| 16 | `Flatten` |
| 10 | `Linear(in_features=16, out_features=10)` |

As show in Table 8, SNE is always either the best model or the second best model in the cross-dataset task. SNE is particularly good in the A→B performance prediction task compared to the next competitive baselines, NFN$_{NP}$ and NFN$_{HNP}$. The MLP baseline, as expected, performs the worse since concatenating all weight values in a single vector looses information such as the network structure. STATNN [Unterthiner et al., 2020] performs relatively better than the MLP baseline suggesting that the statistics of each layer indeed captures enough information to do moderately well on the neural network performance prediction task. NFN$_{NP}$ and NFN$_{HNP}$ perform much better than STATNN and MLP and NFN$_{HNP}$ in particular shows good results in the A→A setting. Interestingly, NFN$_{NP}$ is a much better cross-dataset performance prediction model than NFN$_{HNP}$. However, across the entire cross-dataset neural network performance prediction task, SNE outperforms all the baselines as shown in the last row of Table 8.

### F.2 Cross-Architecture Evaluation

Next we reverse the transfer direction from Arch$_2$ to Arch$_1$ in Section 4.2 and provide the results for SNE in Table 9. Note that for this task NeuralGraph [Kofinas et al., 2024] cannot be used since we transfer from a smaller architecture to a larger one. As can be seen from Table 9, SNE transfers well across architectures further validating the results in Section 4.2.

## G   Architectures for Generating model zoos

We specify the architectures for generating the model zoos of Arch$_1$ and Arch$_2$. For the model zoos of Arch$_1$ in Table 10, all datasets with with 3 channel images (CIFAR10 and SVHN) are converted to grayscale. This is in accordance with the setups of Unterthiner et al. [2020], Navon et al. [2023] and Zhou et al. [2023a,b] and allows us to evaluate both methods in the cross-dataset task for this set of homogeneous model zoos. For model zoos of Arch$_2$ in Tables 11 & 12, we maintain the original channels of the datasets. The cross-architecture transfer task is from Arch$_1$ to Arch$_2$. Note also that for the same dataset, *i.e.*, CIFAR10, the cross-architecture evaluation is also from models trained grayscale to RGB images. All model zoos were generated using the procedure outlined in the Appendix of Unterthiner et al. [2020]. The architecture for the INR experiments is outlined in Table 16.

Table 11: Arch₂ for MNIST.

| Output Size | Layers |
|---|---|
| $1 \times 28 \times 28$ | `Input Image` |
| $8 \times 24 \times 24$ | `Conv2d(in_channels=1 , out_channels=8, kernel_size=5)` |
| $8 \times 12 \times 12$ | `MaxPool2d(kernel_size=2, stride=2), ReLU` |
| $6 \times 8 \times 8$ | `Conv2d(in_channels=8 , out_channels=6, kernel_size=5)` |
| $6 \times 4 \times 4$ | `MaxPool2d(kernel_size=2, stride=2), ReLU` |
| $4 \times 3 \times 3$ | `Conv2d(in_channels=6 , out_channels=4, kernel_size=2), ReLU` |
| 36 | `Flatten` |
| 20 | `Linear(in_features=36, out_features=20), ReLU` |
| 10 | `Linear(in_features=20, out_features=10)` |

Table 12: Arch₂ for CIFAR10 and SVHN.

| Output Size | Layers |
|---|---|
| $3 \times 28 \times 28$ | `Input Image` |
| $8 \times 24 \times 24$ | `Conv2d(in_channels=3 , out_channels=8, kernel_size=5)` |
| $8 \times 12 \times 12$ | `MaxPool2d(kernel_size=2, stride=2), ReLU` |
| $6 \times 8 \times 8$ | `Conv2d(in_channels=8 , out_channels=6, kernel_size=5)` |
| $6 \times 4 \times 4$ | `MaxPool2d(kernel_size=2, stride=2), ReLU` |
| $4 \times 3 \times 3$ | `Conv2d(in_channels=6 , out_channels=4, kernel_size=2), ReLU` |
| 36 | `Flatten` |
| 20 | `Linear(in_features=36, out_features=20), ReLU` |
| 10 | `Linear(in_features=20, out_features=10)` |

## H Dataset Details

Dataset splits for model zoos of Arch₁ is given in Table 13. For the cross-architecture task, we generate model zoos of with 750 neural networks of Arch₂ for testing.

Table 13: Dataset splits for model zoos of Arch₁.

| Model zoo | Train set | Validation set | Test set |
|---|---|---|---|
| MNIST | 11998 | 3000 | 14999 |
| FashionMNIST | 12000 | 3000 | 15000 |
| CIFAR10 | 12000 | 3000 | 15000 |
| SVHN | 11995 | 2999 | 14994 |

## I Hyperparameters

We elaborate all the hyperparameters used for all experiments in Table 14.

Table 14: Hyperparameters for CNN experiments.

| Hyperparameter | Value |
|---|---|
| LR | $1e - 4$ |
| Optimizer | Adam |
| Scheduler | Multistep |
| Batchsize | 64 |
| Epochs | 300 |
| Metric | Binary Cross Entropy |
| SAB Hidden Size | 64 |
| PMA Seed Size | 64 |
| # SAB Blocks | 2 |
| chunksize | 32 |
| SAB LayerNorm | False |

Table 15: Generalization Performance Predictor.

| Output Size | Layers |
|---|---|
| 1000 | `Linear(in_features=1024, out_features=1000), ReLU` |
| 1000 | `Linear(in_features=1000, out_features=1000), ReLU` |
| 1 | `Linear(in_features=1000, out_features=1), Sigmoid` |

Table 16: INR Architecture. Activations are Sinusoidal

| Output Size | Layers |
|---|---|
| 32 | `Linear(in_features=1, out_features=32), Sine` |
| 32 | `Linear(in_features=32, out_features=32), Sine` |
| 1 | `Linear(in_features=32, out_features=1)` |

## J    Implementation Details

SNE is implemented using Pytorch [Paszke et al., 2019]. The SNE model consists of 4 sub-modules:

- **Layer Chunk Encoder:** This consists of two SAB modules where each SAB module is as stack of two SAB layers, followed by a single PMA layer. The layer chunk encoder encodes all the chunks of a given layer independently.
- **Layer Encoder:** This module encodes all the encoded chunks of a layer and consists of two SAB modules and a single PMA layer.
- **Separated Layer Encoder:** This module encodes all the encodings of a layer, for instance the weights and biases, into a single layer encoding vector. It also consists of two SAB modules and a single PMA layer.
- **NN Encoding Layer:** This module takes as input all the layer encodings and compresses them to obtain the neural network encoding which is used for the downstream task of predicting the neural network generalization performance. It also consists of two SAB modules and a single PMA layer.

In addition to the sub-modules above, the layer/level positional encoders are applied to each sub-module when required (see Section 3). The neural network performance predictor, which takes as input the neural network encoding vector from SNE and predicts the performance is detailed in Table 15.

## K    Miscellanea

In Figures 4, 5 and  6 we provide enlarged versions of the figures in Figure 2 for better viewing.

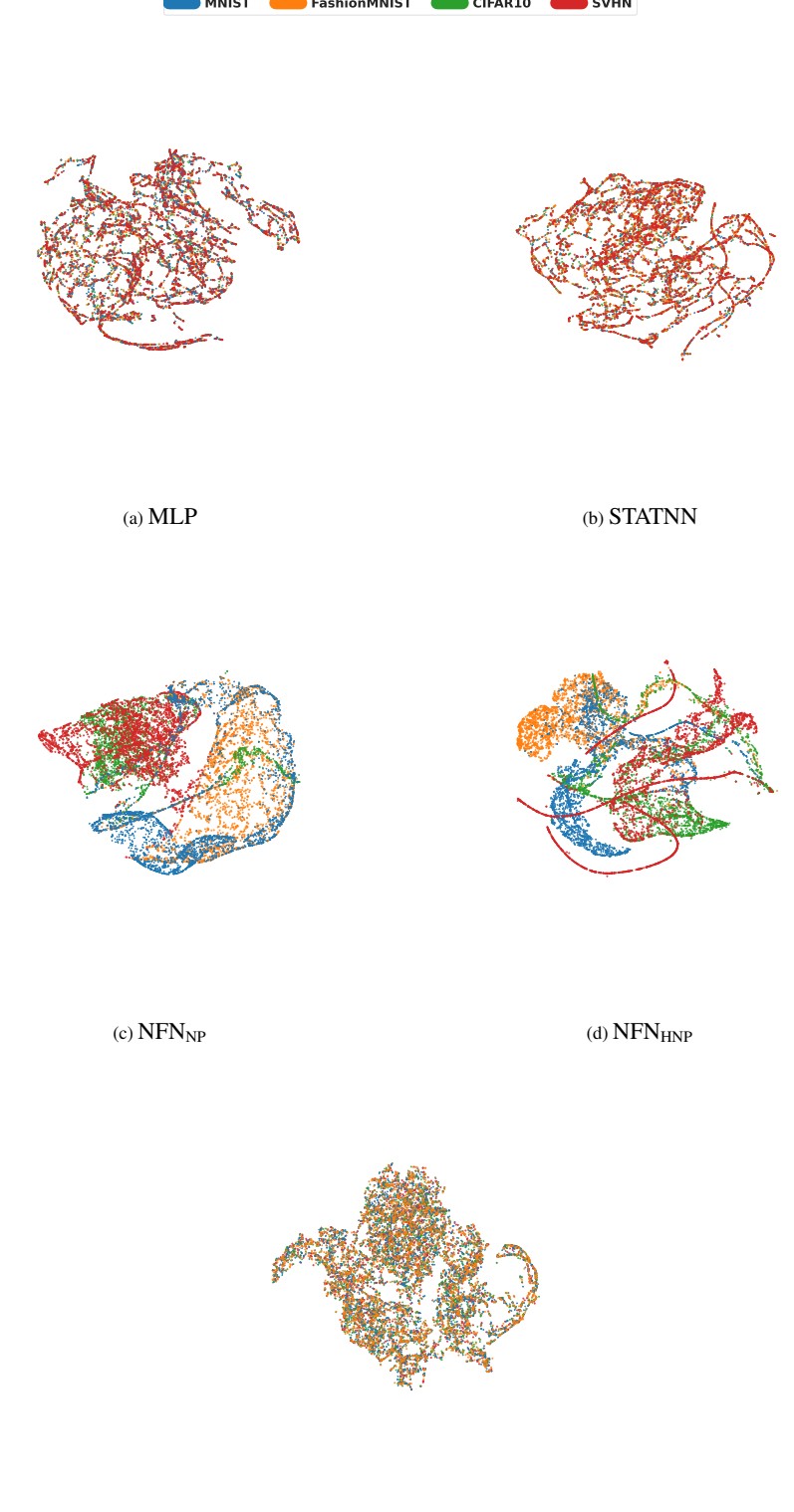

(a) MLP

(b) STATNN

(c) NFN$_{NP}$

(d) NFN$_{HNP}$

(e) SNE(Ours)

Figure 4: TSNE Visualization of Neural Network Encoding. We train neural network performance prediction methods on a combination of the MNIST, FashionMNIST, CIFAR10 and SVHN modelzoos of Unterthiner et al. [2020]. Zoom in for better viewing.

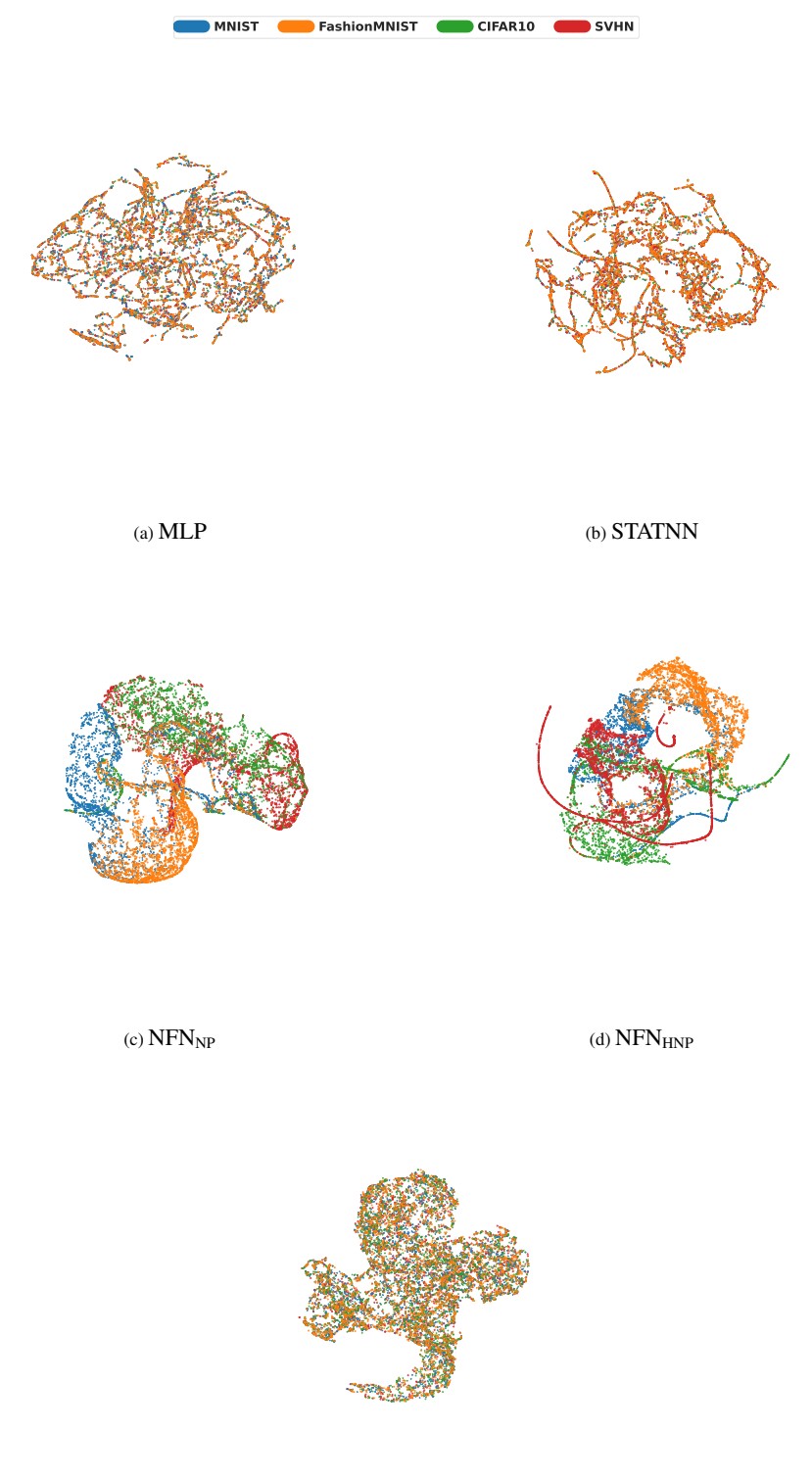

(a) MLP

(b) STATNN

(c) NFN$_{NP}$

(d) NFN$_{HNP}$

(e) SNE(Ours)

Figure 5: TSNE Visualization of Neural Network Encoding. We train neural network performance prediction methods on a combination of the MNIST, FashionMNIST, CIFAR10 and SVHN modelzoos of Unterthiner et al. [2020]. Zoom in for better viewing.

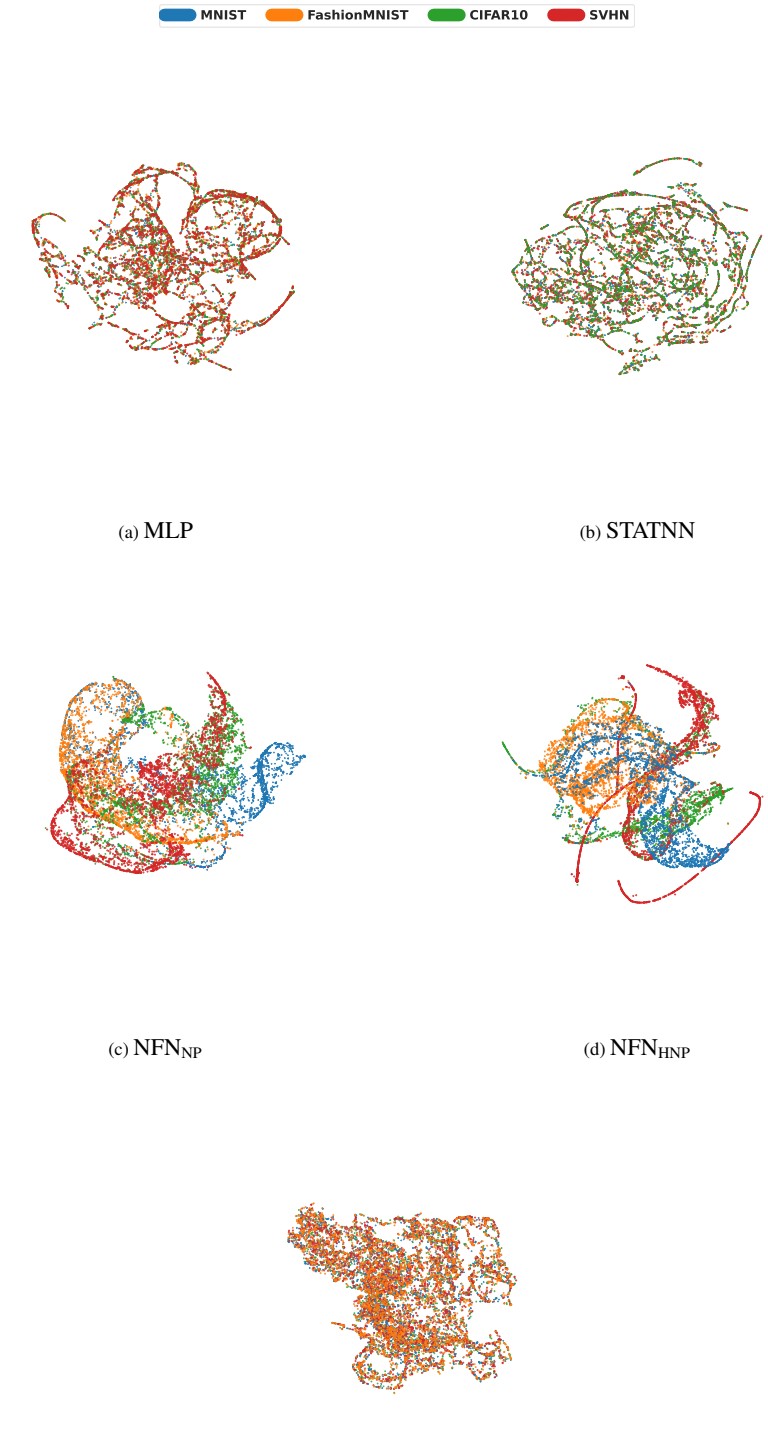

(a) MLP

(b) STATNN

(c) NFN$_{NP}$

(d) NFN$_{HNP}$

(e) SNE(Ours)

Figure 6: TSNE Visualization of Neural Network Encoding. We train neural network performance prediction methods on a combination of the MNIST, FashionMNIST, CIFAR10 and SVHN modelzoos of Unterthiner et al. [2020]. Zoom in for better viewing.

