# OpenReview forum: "Set-based Neural Network Encoding Without Weight Tying"
_NeurIPS.cc/2024/Conference — NeurIPS 2024 poster_

### Official Review · Reviewer_uB2K · 2024-06-13

**Soundness:** 4
**Presentation:** 2
**Contribution:** 3
**Rating:** 6
**Confidence:** 4

**Summary:**

This paper proposes SNE, or Set-based Network Encoding, a method of encoding the weights of arbitrary neural networks in order to predict properties such as performance, generalization gap, training loss, number of epochs, etc.  Specifically, using a chunking mechanism as well as layer-wise and block-wise positional encodings to generalize across different types of neural networks. Transferability experiments are conducted where a predictor is trained on one network model zoo, then evaluate on another, demonstrating an ability few predictors have. Additional experiments and ablations are performed, producing convincing results. Finally, the paper is decently well-written, but has a few fixable presentation flaws.

**Strengths:**

- The contribution of the paper, using a network's weights to determine attributes about it, are quite novel.
- The Methodology section is quite clear and easy to follow. Specifically, the aggregation mechanism in Eq. 9 is similarly analogous to graph aggregation in GNNs.
- The experimental results are generally convincing.
- Some visualization of the network representations from different methods are provided.
- There are ablation results included in the appendix.
- A limitations section is provided, highlighting reasonable restrictions on the experiments.

**Weaknesses:**

- First, the motivation of this paper, specifically lines 29-30, is quite weak and fails to really provide examples of the application of this research other than asserting that they must exist.
- Overall, the presentation is a weak spot, and should be revised. Some examples:
	- E.g., in line 31, the authors mention "Implicit Neural Representations (INR)" - what are these? No citation is provided. By contrast, "b) the performance of CNNs" is very easy to understand.
	- "modelzoo", this should be "model-zoo" [1] or "model zoo" [2-4].
	- Citations in the paper have issues. For NeurIPS its generally just [num], or if you want to be fancier, "Author et al., Year [num]" but not "Author et al. [num]" without the year.
	- Float captions need work, e.g., Tabs 2/3 need longer captions to fully explain the performance metric Kendall's Tau, etc. Should be longer than 1 sentence.
	- Figure 2 needs work. The plots should be bigger and legend is clipped.
	- NeurIPS checklist should be placed after the references and supplementary.
- The evaluation is done on simpler datasets/networks, however, the limitations section highlights why.

**Questions:**

I find the idea of predicting end-to-end neural network attributes from the weight values to be quite fascinating. It is very similar to the concept of a neural predictor [5] in Neural Architecture Search [6] but quite orthogonal in the approach, relying on the weights instead of the graph structure. Similarly, most neural predictors suffer from a lack of generalizability/transferability [7, 8], like most of the baselines in this paper. Can the authors provide some commentary/comparison between these two umbrella fields?

Similarly, is it possible to leverage the encoding mechanism (Eqs. 1-9) to derive which types of weights/layers/values are responsible for certain end-to-end attributes, e.g., performance of epochs, e.g., as [9] do?

Refs:

[1] "Equivariant Architectures for Learning in Deep Weight Spaces" - ICML'23.

[2] https://modelzoo.co/

[3] https://github.com/onnx/models

[4] https://pytorch.org/serve/model_zoo.html

[5] "How Powerful are Performance Predictors in Neural Architecture Search?" - NeurIPS'21.

[6] "Efficient Neural Architecture Design via Capturing Architecture-Performance Joint Distribution" - AISTATS'24.

[7] "GENNAPE: Towards Generalized Neural Architecture Performance Estimators" - AAAI-23.

[8] "Bridge the Gap Between Architecture Spaces via A Cross-Domain Predictor" - NeurIPS'22.

[9] "Building Optimal Neural Architectures using Interpretable Knowledge" - CVPR'24.

**Limitations:**

Yes, limitations and future work is discussed in App. B. The authors explain that their experiments are on simpler/smaller/older networks as they have limited computational resources. I find this to be an acceptable justification.

---

> ### Author Rebuttal · Authors · 2024-08-07
>
> We thank the Reviewer for taking the time to offer constructive feedback for improving the paper. We respond to the questions raised below.
>
> **Provide examples of the application of this research other than asserting that they must exist.**
>
> We add the following to elaborate on the potential applications of this research to complement lines 29-30:
> ‘The ability to infer such fundamental properties of trained neural networks using only the parameter values has the potential to open up new application and research paradigms such as  learning in the latent space of neural network weights for tasks such as weight generation [1, 2], latent space transfer of weights across datasets allowing for transferring weights from one dataset to another as was recently demonstrated in [3] and latent space optimization using gradient descent where optimization is performed on the weight embeddings [4].’
> We hope this addition enhances the motivation for this line of research.
>
> **Overall, the presentation is a weak spot, and should be revised. Some examples:**
> - We have added a citation to Implicit Neural Representations (INR) [1] and added the following description, “INRs are neural parameterizations of signals such as images using multi-layer perceptrons”.
> - All occurrences of ‘modelzoo’ have been replaced with ‘model zoo’.
> - We have changed the citation format to ‘Author et al., Year[num]’.
> - We’ve included the following to captions on Tables 2/3 “Models are evaluated using Kendall’s Tau[5], a rank correlation metric.’ with a citation to the paper that introduces the metric.
> - We have reduced the number of plots in Figure 2 to one per model and increased the size (see Figure 2 in the attached pdf). The remaining figures are moved to the appendix in enlarged form and the clipped legend fixed.
> - The checklist has been placed after the references and supplementary as pointed out.
>
> **The evaluation is done on simpler datasets/networks, however, the limitations section highlights why.**
>
> Based on the suggestion of Reviewer nYWS we provide additional results on more complicated networks, vision transformers, in our response to nYWS and in the general response. We hope that these additional results on more complicated architecture demonstrates the utility of the proposed method.
>
> **Can the authors provide some commentary/comparison between these two umbrella fields (NAS and weight encoding)?**
>
> In NAS, the predictors mostly rely on the computation graph of generated architectures. This is quite different from the problem we consider where we require access to the parameter values themselves, optimized for some number of epochs. However, the general underlying problem has commonality between the two fields: namely predicting some properties of the networks in consideration. Perhaps a new line of exploration would consider combining both fields. For instance, a neural network encoder, like ours, could be trained to conditionally take as input either the weights, the computational graph or both. Such a pre-trained predictor could be applied to NAS where in test mode, only the computational graph is given to estimate the performance of newly generated architectures. This would require the encoder to be agnostic to architectural choices like our model. The discovered architectures when trained could then be used to update the property predictor using both weights and computation graphs. Such a combination would find application in both fields. To the best of our knowledge, we are unaware of any works that attempt such a combination and we leave this exploration as future work and will draw attention to such a merger of both fields in our discussion and limitation sections.
>
> **Similarly, is it possible to leverage the encoding mechanism (Eqs. 1-9) to derive which types of weights/layers/values are responsible for certain end-to-end attributes, e.g., performance of epochs, e.g., as [9] do?**
>
> Technically, such an analysis should be possible. For instance, we could train SNE to predict multiple properties for a given network and at test time, we can remove the encodings of different layers (in Eq 9) and see how it affects the predictions compared to the full network. This will provide a mechanism for determining which layers/weights are responsible for certain attributes. Currently we are unable to do such an analysis since all the model zoos used in this paper only come with a single attribute in the meta-data. Such analysis will require regenerating the model zoos, a task which we are unable to undertake during the short rebuttal period. However, we are inclined to agree with the Reviewer that such an analysis will be very useful.
>
> **References**
>
> [1] Hyper-Representations as Generative Models: Sampling Unseen Neural Network Weights, 2022
>
> [2] Wang, Kai, et al. "Neural Network Diffusion." arXiv preprint arXiv:2402.13144 (2024).
>
> [3] Soro, Bedionita, et al. "Diffusion-based Neural Network Weights Generation." arXiv preprint arXiv:2402.18153 (2024).
>
> [4] Rusu, Andrei A., et al. "Meta-learning with latent embedding optimization." arXiv preprint arXiv:1807.05960 (2018).
>
> [5] Kendall, M. G. A new measure of rank correlation. Biometrika, 30(1/2):81–93, 1938.

---

> ### Comment · Reviewer_uB2K · 2024-08-09
> **5->6**
>
> I thank the authors for their detailed rebuttal, which satisfactorily addressed my concerns and questions. After reading the other reviewer comments and corresponding rebuttals I find that the reasons to accept outweigh concerns to reject, and thus raise my score. If accepted, I encourage the authors to integrate rebuttal experimental results as well as revise related work and limitations to reflect these past discussions.

---

> > ### Author Response · Authors · 2024-08-10
> > **Discussion**
> >
> > We would like to thank the Reviewer for the continued engagement. As requested, the additional experimental results as well as the revised related work, limitations and discussions of potential applications and similarities to NAS research will be incorporated in the paper. Thank you.

---

### Official Review · Reviewer_EgRw · 2024-07-11

**Soundness:** 3
**Presentation:** 2
**Contribution:** 3
**Rating:** 7
**Confidence:** 4

**Summary:**

This work introduces a set-based neural network encoding that processes each chunk of an input model independently to predict network properties. The proposed model, SNE, is trained using Logit Invariance instead of weight tying to maintain generalizability to unseen architectures. Unlike previous approaches, SNE can be trained and tested on different architectures and it demonstrated superior performance in out-of-distribution experiments involving both datasets and architectures.

**Strengths:**

SNE is more general than its predecessors and can be applied to any input architecture. Results indicate that it outperforms other encoding methods.

**Weaknesses:**

While SNE shows improvements in novel tasks such as cross-architecture and cross-dataset scenarios, its comparison on traditional tasks is limited to predicting INR frequencies. Expanding Table 1 to include SNE's performance on additional outputs, such as accuracy, would enhance the paper.

**Questions:**

The writing is generally good but could be improved in certain areas. For example, the abstract contains a redundant sentence: “by utilizing a layer-wise encoding scheme that culminates to encoding all layer-wise encodings to obtain the neural network encoding vector,” which should be revised.

**Limitations:**

Limitations were properly addressed in appendix.

---

> ### Author Rebuttal · Authors · 2024-08-07
>
> We thank the Reviewer for taking the time to offer constructive feedback for improving the paper. We respond to the questions raised below.
>
> **The writing is generally good but could be improved in certain areas. For example, the abstract contains a redundant sentence: “by utilizing a layer-wise encoding scheme that culminates to encoding all layer-wise encodings to obtain the neural network encoding vector,” which should be revised.**
>
> We have removed the sentence from the abstract and replaced it with ‘Furthermore, our $\textbf{S}$et-based $\textbf{N}$eural network $\textbf{E}$ncoder (SNE) takes into consideration the hierarchical computational structure of neural networks.’ We hope this resolves the redundancy pointed out.
>
> **Expanding Table 1 to include SNE's performance on additional outputs**
>
> Our request to [1] for the dataset of INRs for the ShapeNet dataset is currently pending and this task will be added to Table 1 when the dataset is received. However we note that fundamentally, there is no difference between the accuracy prediction task for INRs and the frequency prediction task that we present in Table 1 since these are all evaluations on MLPs.
>
> We would also like to draw the Reviewers attention to the additional results we provide on generalization on vision transformers which we provide in our general response. We hope these additional evaluations further demonstrate the wide range applicability of our method.
>
> **References**
>
> [1] De Luigi, Luca, et al. "Deep learning on implicit neural representations of shapes." arXiv preprint arXiv:2302.05438 (2023).

---

### Official Review · Reviewer_nYWS · 2024-07-12

**Soundness:** 3
**Presentation:** 3
**Contribution:** 3
**Rating:** 6
**Confidence:** 3

**Summary:**

This work tackles an original and interesting challenge: predicting neural networks properties from their trained weight values. To do so, the authors propose to leverage set to set and set to vector transformations in order to encode weight values. Furthermore, they propose to account for the deep neural network architecture and operation order through the addition of positional encoding at several stages of the property predicting model. Ultimately, they anticipate the problem of various model sizes with a padding/chinking mechanism.
The proposed method is then evaluated on small scale datasets in a cross-dataset fashion against method of the field, published in recent years.

**Strengths:**

Such research studies can have a significant in other fields. For instance, a good accuracy estimator from model weight values can be leveraged as a proxy objective in DNN quantization or weight pruning.
The approach is well motivated and the solutions to the highlighted challenges are sound.
The paper is well written and simple to follow.
The authors provide a thorough evaluation and ablation study of the proposed method and its components.

**Weaknesses:**

From my understanding of this work, I identified two major drawbacks:
1. the authors insist on the generalization across different model architectures and tasks. From my perspective, the former is the most important one. However, as it stands, there is little variation between Arch1 and Arch2 as presented in this study: both are simple CNNs. I wonder if the authors could provide numbers w.r.t transformer architectures against CNNs.
2. Since these experiments can be expensive (require multiple training of DNNs to create the predictor training set), it would be interesting to measure the impact of the dataset size w.r.t the predictor's performance, in order to estimate how far the community is from being to reasonably work on large scale datasets such as ImageNet.

As it stands, I believe this work aims at providing empirical results which are not fully convincing yet. I look forward to the authors' response and other reviews and will update my rating accordingly.

**Questions:**

I summarize the previous points:
1. could the authors provide results with a predictor trained on CNNs from arch1 and predict the performance of transformers?
2. could the author measure the importance of the training set size (pairs of DNNs and accuracy) w.r.t. the predictor's performance, in order to estimate the cost of working on ImageNet-like tasks?

**Limitations:**

The authors made a relevant attempt at highlighting the limitations of their work

---

> ### Author Rebuttal · Authors · 2024-08-07
>
> We thank the Reviewer for taking the time to offer constructive feedback for improving the paper. We respond to the questions raised below.
>
> **could the authors provide results with a predictor trained on CNNs from arch1 and predict the performance of transformers?**
>
> We provide the requested results below. We generate a model zoo of transformer classifiers based on the [1] and test the transfer from arch1 to the transformer model zoo as requested.
> For this task, we are unable to benchmark against NeuralGraph[2] as was done in Table 2, since the model cannot process transformer weights when trained on Arch1. Hence we benchmark against the DeepSets baseline as in Table 2. We present the results in the table below:
>
> | Arch1 $\rightarrow$ Transfer       	| DeepSets | SNE(Ours)  |
> |----------------------------|----------|------|
> | MNIST $\rightarrow$ MNIST	| 0.1975 $\pm$ 0.000 	| **0.4625** $\pm$ 0.006 |
> | CIFAR10 $\rightarrow$ MNIST   | 0.1970 $\pm$ 0.000 	| **0.3278** $\pm$ 0.029 |
> | SVHN $\rightarrow$ MNIST | 0.1906 $\pm$ 0.000 	| **0.3735** $\pm$ 0.009 |
>
> From the results, the proposed SNE generalizes better to the unseen transformer architecture at test time than the baselines showing strong architectural transfer. Additionally, here, the model encodes an architecture with about 5 times the number of parameters in Arch1 demonstrating the scalability of our approach. The DeepSets baseline fails to generalize on this task.
>
> **could the author measure the importance of the training set size w.r.t. the predictor's performance, in order to estimate the cost of working on ImageNet-like tasks?**
>
> We provide the requested results in Figure 1 of the attached pdf. From this it can be seen that SNE is more data efficient compared to the baselines (DWSNet[2] and NFT[3]) across all percentages of the full training data demonstrating that the proposed method learns a good embedding even in the limited data setting. We thank the Reviewer for suggesting this ablation as it further demonstrates the strength of SNE.
>
>
> **References**
>
> [1] https://github.com/s-chh/PyTorch-Scratch-Vision-Transformer-ViT
>
> [2] Aviv Navon, Aviv Shamsian, Idan Achituve, Ethan Fetaya, Gal Chechik, and Haggai Maron. Equivariant architectures for learning in deep weight spaces. arXiv preprint arXiv:2301.12780, 2023.
>
> [3] Allan Zhou, Kaien Yang, Yiding Jiang, Kaylee Burns, Winnie Xu, Samuel Sokota, J Zico Kolter, and Chelsea Finn. Neural functional transformers. arXiv preprint arXiv:2305.13546, 2023.

---

> > ### Comment · Reviewer_nYWS · 2024-08-08
> > **Response to rebuttal**
> >
> > I would like to thank the authors for their response and the new experimental results they provided for this rebuttal.
> > Overall, I consider that my concerns have been addressed. Thus, I am likely to increase my rating for this work after the end of the discussion period.

---

### Official Review · Reviewer_bv1s · 2024-07-13

**Soundness:** 2
**Presentation:** 2
**Contribution:** 2
**Rating:** 4
**Confidence:** 4

**Summary:**

The study introduces Set-based Neural Network Encoding (SNE) without weight tying, allowing for encoding network information into a compact form. SNE utilizes Logit Invariance Regularization to ensure the correct functional equivalence of network permutations, departing from traditional weight tying methods. The choice of Set-to-Set and Set-to-Vector functions, including attention mechanisms and pooling layers, are crucial in implementing the SNE framework. Experimental results demonstrate the effectiveness of SNE in predicting generalization performance for Convolutional Neural Networks (CNNs) and Incremental Neural Networks (INRs). SNE is compared with various baselines such as MLP, Deepsets, HyperRep, and others, showcasing its superiority in certain aspects.

**Strengths:**

- New evaluation benchmarks of cross-dataset and cross-architecture evaluation.

- Solid empirical performance. SNE significantly outperforms HyperRep by very large margins in the cross-architecture task. Also, SNE demonstrates true agnosticism to architectural choices compared to NeuralGraph, and shows robust performance without being trained on the training set of a specific model zoo.

**Weaknesses:**

- The proposed method did not ensure weight-space equivariance by design but use a way that is more like an augmentation plus a consistency loss. The theoretic contribution is therefore a bit weak.

- The padding and chunking designs look tricky and it is not clear for me how they help with the performance.

- Fig 1 is not really clear. (1) the legend is not sufficient to understand the computation diagram presented in the figure. (2) the input is shown as neurons but from the method I think it should be the weight matrix that connecting neurons in adjacent layers.

**Questions:**

- In table 1 NeuralGraph is not reported. Is it because the method is not applicable there?

- Does the method need more optimization steps than baseline methods that ensure weight-space symmetry by design? I suppose optimization would be harder because it requires to optimize for the logit invariance.

**Limitations:**

The limitations of the method are discussed in Appendix B

---

> ### Author Rebuttal · Authors · 2024-08-07
>
> We thank the Reviewer for taking the time to offer constructive feedback for improving the paper. We respond to the questions raised below.
>
>
> **The proposed method did not ensure weight-space equivariance by design but use a way that is more like an augmentation plus a consistency loss. The theoretic contribution is therefore a bit weak.**
>
>
> While we use a regularization approach to achieve approximate minimal equivariance in weight-space, our usage of the logit invariance regularizer [1] theoretically guarantees that we indeed learn the correct invariance property similar to the weight-typing approaches used by the baselines. Additionally, our formulation is what allows us to deal with arbitrary architectures using a single model, as opposed to the baselines, since strict enforcement of weight-space equivariance by design requires crafting a new model for every distinct architecture. In this sense, our approach provides a general encoder which in principle is applicable to any architecture, resolving the limitations of purely weight-tying approaches.
>
> **The padding and chunking designs look tricky and it is not clear for me how they help with the performance.**
>
>
> The padding and chunking operations are introduced in our pipeline to offer computational efficiency in cases where the parameters are large and hence infeasible to process all together. In Table 6 of the appendix we provide an ablation on the effect of the chunk size, which we reproduce below on the frequency prediction task:
>
> | ChunkSize | MSE                 |
> |-----------|---------------------|
> | 4         | $0.095 \pm 0.012$ |
> | 8         | $0.090 \pm 0.023$ |
> | 16        | $0.118 \pm 0.019$ |
> | 32        | $0.056 \pm 0.020$ |
>
> Here we see that performance stays almost the same until the largest chunksize (which in this case is set to the largest size hence requiring no chunk/pad operations). The chunksize is selected to suit the gpu memory requirements allowing us to train our method even with limited gpu memory which in our case is a single GTX 1080 gpu with 11GB of memory.
>
>
> **Fig 1 is not really clear. (1) the legend is not sufficient to understand the computation diagram presented in the figure. (2) the input is shown as neurons but from the method I think it should be the weight matrix that connecting neurons in adjacent layers.**
>
>
>
> We have modified the figure to show the weight matrix instead of the layer in compressed form as shown in the figure in the updated paper. We thank the Reviewer for pointing this out as this improves the clarity of the presented pipeline. Additionally, we link the relevant approach subsections to each stage in the figure to aid clarification of the computation diagram.
>
> **In table 1 NeuralGraph is not reported. Is it because the method is not applicable there?**
>
> Yes. We were unable to apply NeuralGraph for this task. Before submission, correspondences with the authors of NeuralGraph could not resolve the issue we encountered when applying the method for this task and that of Table 4.
>
>
> **Does the method need more optimization steps than baseline methods that ensure weight-space symmetry by design?**
>
> No. In all experiments, we train our method for the same number of epochs as all the baselines. The addition of the logit invariance regularizer introduces no further optimization steps in all our experiments and we encountered no difficulties in minimizing the general objective together with the logit invariance loss.
>
> **References**
>
> [1] Artem Moskalev, Anna Sepliarskaia, Erik J Bekkers, and Arnold WM Smeulders. On genuine invariance learning without weight-tying. In Topological, Algebraic and Geometric Learning Workshops 2023, pages 218–227. PMLR, 2023

---

> > ### Comment · Reviewer_bv1s · 2024-08-08
> >
> > Thanks for the response.
> >
> > - I wonder what the issue was with NeuralGraph on the experiments in Table 1 and Table 4? If the NNs for the INRs are MLP architecture, I do not expect any issue there since NeuralGraph is able to work on classification of MNIST / CIFAR in INRs.
> >
> > - The experiments on INRs now are on frequency predictions of INRs fitting simple sine waves, which is pretty low-level and toy-ish. I have the concerns about not following the conventions in previous weight space symmetry methods (DWSNet, NFT, NeuralGraphs, etc.) to test INR classification tasks on MNIST, FashionMNIST and CIFAR. This should be a more high-level and sophisticated task than sine wave frequencies, and direct comparison with performance reported by prior work would be more convincing.
> >
> > - What is the NN architecture tested for the chunksize experiments?

---

> ### Author Response · Authors · 2024-08-10
> **Discussion**
>
> We would like to thank the Reviewer again and respond to the new queries below:
>
> **I wonder what the issue was with NeuralGraph on the experiments in Table 1 and Table 4? If the NNs for the INRs are MLP architecture, I do not expect any issue there since NeuralGraph is able to work on classification of MNIST / CIFAR in INRs.**
>
>
> - Firstly, we utilize the official implementation of NeuralGraphs provided at [1].
> In Table 1 we are unable to get NeuralGraphs to converge on this task using multiple hyper-parameter runs.
> Secondly, the experiments of Table 4 are on ConvNets (the architecture is outlined in Tables 10 and 11 of the appendix. In Table 4, we evaluate in terms of cross-dataset performance prediction and the issue with NeuralGraph, under this evaluation setting, lies in the following: we are unable to obtain positive correlation coefficients when we evaluate it cross-dataset. As we elaborated in our response to Reviewer bv1s,  in our correspondences with the authors of NeuralGraph, we were unable to find and resolve the issues with negative correlation values in Table 4. Interestingly, we do not observe these issues in our evaluation of NeuralGraph in Table 2.
> These explain the absence of NeuralGraphs in Tables 1 and 4.
>
>
> **This should be a more high-level and sophisticated task than sine wave frequencies, and direct comparison with performance reported by prior work would be more convincing.**
>
> - As we elaborated in our response to Reviewer EgRw, there is no fundamental difference between the task of INR frequency prediction and INR classification as these require the same guarantees in MLP weight space. Our request to [1] for the dataset of INRs for the ShapeNet dataset is currently pending and this task will be added to Table 1 when the dataset is received together with INR classification (a task which we are currently undertaking but do not expect to be completed during this discussion period. This is partly due to the large datasets required for for training INR classification for MNIST, FashionMNIST and CIFAR: all of DWSNet, NFT, NeuralGraph generate up to a million datapoints as opposed to the 60000 samples in the original MNIST classification task, resulting in long training iterations).
> Importantly, we point the Reviewers attention to the following: that all the experiments presented in Table 2, 3,4 and 7  of the appendix together with the newly introduced Transformer evaluation task suggested by Reviewer NYWS are much more difficult tasks compared to classifying INRs since these  require respecting symmetric properties in both CNN (with MLP heads) and Transformer weight spaces. We hope this provides some clarity.
>
> **What is the NN architecture tested for the chunksize experiments?**
>
> - We utilize the architecture outlined in Table 15 of the appendix which consists of 3 linear layers each with hidden size of 32 and the first two followed by sinusoidal activation function. This is the architecture for generating the model zoo used in the experiments of Table 1.
>
>
> Finally, we would like to thank the Reviewer for the continued engagement as both the discussions and our response to the original review concerns as well as the questions we’ve presently addressed greatly improves clarity of the paper and these will be incorporated in the paper.
>
> **References**
>
> [1]https://github.com/mkofinas/neural-graphs/tree/1f2b671ab4988ef212469363005a5b99eec16580
>
> [2] De Luigi, Luca, et al. "Deep learning on implicit neural representations of shapes." arXiv preprint arXiv:2302.05438 (2023).

---

> > ### Author Response · Authors · 2024-08-12
> >
> > Dear Reviewer,
> > We thank you for taking the time to provide feedback. As the discussion period draws to an end, we would like to know if there are any further queries you'd like as to respond to.
> >
> > Thank you for your time.

---

> > > ### Comment · Reviewer_bv1s · 2024-08-12
> > >
> > > Thanks for the response.
> > >
> > > I still have concerns about not directly conducting the experiments on the most typical benchmark (MNIST and CIFAR image INRs) established in many prior papers in this field. From my own experience (and may also be suggested by observations from [A]), if the training of the INRs is harder (takes longer steps of optimization to converge), then it is likely more challenging for downstream weight-space tasks as well. Sine wave INRs are not sufficient as a replacement in my view, because they look too simple and toy-ish. Without checking the performance on the most common benchmark, I am not sure if the evaluation is convincing enough.
> > >
> > > I see the values of this work including the new benchmark. However aggregating the experimental evaluation and the innovations introduced in the method, I tend to keep a borderline score.
> > >
> > > [A] Papa et al. How to Train Neural Field Representations: A Comprehensive Study and Benchmark. In CVPR 2024

---

> > > > ### Author Response · Authors · 2024-08-13
> > > >
> > > > We thank the Reviewer for the feedback.

---

### Author Rebuttal · Authors · 2024-08-07

**General Response**

We thank all Reviewers for taking the time to offer constructive feedback for improving the paper. Based on the suggestions of Reviewers nYWS, we provide the following additional results.


**Evaluation on Transformers**

We generate a model zoo of transformer classifiers based on the [1] and test the transfer from arch1 to the transformer model zoo as requested.
For this task, we are unable to benchmark against NeuralGraph[2] as was done in Table 2, since the model cannot process transformer weights when trained on Arch1. Hence we benchmark against the DeepSets baseline as in Table 2. We present the results in the table below:

| Arch1 $\rightarrow$ Transfer           | DeepSets | SNE(Ours)  |
|----------------------------|----------|------|
| MNIST $\rightarrow$ MNIST    | 0.1975 $\pm$ 0.000     | **0.4625** $\pm$ 0.006 |
| CIFAR10 $\rightarrow$ MNIST   | 0.1970 $\pm$ 0.000     | **0.3278** $\pm$ 0.029 |
| SVHN $\rightarrow$ MNIST | 0.1906 $\pm$ 0.000     | **0.3735** $\pm$ 0.009 |



From the results, the proposed SNE generalizes better to the unseen transformer architecture at test time than the baselines showing strong architectural transfer. Additionally, here, the model encodes an architecture with about 5 times the number of parameters in Arch1 demonstrating the scalability of our approach. The DeepSets baseline fails to generalize on this task.

**Measuring the Importance of training set size.**

We provide a plot of training set size versus error in Figure 1 of the attached pdf. From this it can be seen that SNE is more data efficient compared to the baselines (DWSNet[2] and NFT[3]) across all percentages of the full training data demonstrating that the proposed method learns a good embedding even in the limited data setting. We thank the Reviewer for suggesting this ablation as it further demonstrates the strength of SNE.

We hope that these additional results further emphasizes the benefits of the proposed method.

**References**

[1] https://github.com/s-chh/PyTorch-Scratch-Vision-Transformer-ViT

[2] Aviv Navon, Aviv Shamsian, Idan Achituve, Ethan Fetaya, Gal Chechik, and Haggai Maron. Equivariant architectures for learning in deep weight spaces. arXiv preprint arXiv:2301.12780, 2023.

[3] Allan Zhou, Kaien Yang, Yiding Jiang, Kaylee Burns, Winnie Xu, Samuel Sokota, J Zico Kolter, and Chelsea Finn. Neural functional transformers. arXiv preprint arXiv:2305.13546, 2023.

---

### Decision · Program_Chairs · 2024-09-25

**Decision:**

Accept (poster)

**Comment:**

The paper proposes to predict neural network training properties based on set-based representations of neural network parameters. There exist mixed reviews, but reviewers agree that the approach has novelty. The main complaint is comparisons with priors works including NeuralGraphs, as well as the findings of Papa et al., How to Train Neural Field Representations: A Comprehensive Study and Benchmark. In CVPR 2024. The authors make a fair argument that they are expecting a response from the authors before including the final results. I suggest the paper to be accepted under the condition that the include the more detailed comparisons with NeuralGraph that they currently miss, and discuss their findings compared to Papa et al.